# Preclinical development of kinetin as a safe error-prone SARS-CoV-2 antiviral able to attenuate virus-induced inflammation

Orally available antivirals against severe acute respiratory syndrome coronavirus 2 (SARS-CoV-2) are necessary because of the continuous circulation of new variants that challenge immunized individuals. Because severe COVID-19 is a virus-triggered immune and inflammatory dysfunction, molecules endowed with both antiviral and anti-inflammatory activity are highly desirable. We identified here that kinetin (MB-905) inhibits the in vitro replication of SARS-CoV-2 in human hepatic and pulmonary cell lines. On infected monocytes, MB-905 reduced virus replication, IL-6 and TNFα levels. MB-905 is converted into its triphosphate nucleotide to inhibit viral RNA synthesis and induce error-prone virus replication. Coinhibition of SARS-CoV-2 exonuclease, a proofreading enzyme that corrects erroneously incorporated nucleotides during viral RNA replication, potentiated the inhibitory effect of MB-905. MB-905 shows good oral absorption, its metabolites are stable, achieving long-lasting plasma and lung concentrations, and this drug is not mutagenic nor cardiotoxic in acute and chronic treatments. SARS-CoV-2-infected hACE-mice and hamsters treated with MB-905 show decreased viral replication, lung necrosis, hemorrhage and inflammation. Because kinetin is clinically investigated for a rare genetic disease at regimens beyond the predicted concentrations of antiviral/anti-inflammatory inhibition, our investigation suggests the opportunity for the rapid clinical development of a new antiviral substance for the treatment of COVID-19.

Since the emergence of severe acute respiratory syndrome coronavirus 2 (SARS-CoV-2), the causative agent of 2019 coronavirus disease (COVID-19), global health has been confronted by an unprecedented burden that has taken over 200,000 lives/month in almost two years[1], causing deep economic depression and continued uncertainties. Despite previous threats imposed by highly pathogenic coronaviruses in 2002 and 2014, the response to these diseases did not result in clinically approved vaccines or antivirals. Thus, the development of prophylactic and therapeutic responses to COVID-19 had to unfold during the ongoing pandemic[2].

The unprecedently fast development of vaccines against SARS-CoV-2 has resulted in reduced rates of hospitalization and deaths[3]. However,

SARS-CoV-2 variants of concern (VoC) are constantly emerging with the ability to escape the vaccinal humoral response[4], thus provoking new pandemic waves[5]. VoC emergence highlights that the virus genetic barrier to becoming resistant may not be insurmountable. In fact, antiviral-resistant SARS-CoV-2 strains have been identified to remdesivir and nirmatrelvir[6,7]. This notion reinforces the urgency of antiviral development as an additional alternative to fight COVID-19. Moreover, antivirals are necessary because the sustainability of the anti-SARS-CoV-2 vaccine response may be short and require frequent boosters[8], especially for individuals with an impaired immune system, for whom vaccines may not work properly. Thus, the development of more effective and safe antivirals continues to be an unmet medical need.

✉ e-mail: thiago.moreno@fiocruz.br; joao.calixto@cienp.org.br; jrabi@microbiologica.ind.br

As has been the case with other viral infections, the possibility of inhibiting RNA polymerase using nucleoside/nucleotide analogs has aroused special interest[9,10]. Structural and enzymatic studies indicate that SARS-CoV-2 RNA polymerase [composed of the nonstructural protein 12 (nsp12) and the cofactors nsp7/nsp8] is a key enzymatic system in the virus life cycle[11]. Nsp12 is relatively promiscuous to incorporate the activated forms of various antiviral nucleotides[11]: (i) nonobligate chain terminators, such as remdesivir (RDV), AT-527, and sofosbuvir (SFV); (ii) obligate chain terminators, such as tenofovir (TFV); and (iii) the error-prone purine analogs favipiravir (FPV) and molnupiravir (N4-hydroxycytidine, EIDD-2801, MK-4482). Certain factors may have jeopardized the unequivocal success of these antivirals against COVID-19. The phosphoroamidates[12] RDV, AT-527, and SFV seem to be highly vectorized to the liver, limiting their bioavailability in the lungs. SFV[13] and FPV[14] require higher plasma concentrations against SARS-CoV-2 than their plasma exposures for treating hepatitis C and influenza virus, respectively. The required higher doses of these drugs may be because nsp12 RNA-dependent RNA polymerase (RdRp) activity is coupled with a viral exonuclease (nsp14/10) that proofreads the newly synthetized viral genome and excises atypical nucleotides[15,16]. Molnupiravir, on the other hand, delivers a tautomeric N4-hydroxycytidine in the viral RNA, leading to error-prone catastrophic replication beyond the excision capacity of the exonuclease[17]. Despite relevant concerns that N4-hydroxycytidine may lead to mutagenesis in the host cells[18,19], clinical trials with molnupiravir have demonstrated that treated patients undergo faster SARS-CoV-2 RNA clearance[20] and reduced hospitalization rates[21]. Molnupiravir was not particularly effective in hospitalized individuals[21]. Nevertheless, it was authorized by the Food and Drug Administration (FDA), with a narrow difference of votes (https://www.youtube.com/watch?v = fR9FNSJT64M)[22], and other regulatory agencies in different countries for early antiviral intervention. Furthermore, in July 2022, the World Health Organization (WHO) guidelines for therapeutics and COVID-19 categorized molnupiravir as having weak recommendations in favor[23].

Indeed, in hospitalized COVID-19 patients, especially critically ill patients, SARS-CoV-2 triggers an unbalanced host-dependent response involving a proinflammatory cytokine storm and coagulopathy[24,25]. The progression from mild to severe COVID-19 is characterized by intense viral replication and tissue necrosis, with release of lactate dehydrogenase (LDH) in the respiratory tract[24], involvement of possible neurogenic spread of the virus[26–28] and leukopenia[24,25]. In this harmful scenario, the increased levels of IL-6, TNFα, and C-reactive protein, among other proinflammatory substances, result in a virus-triggered sepsis-like disease. Anti-inflammatory and anti-clotting agents enhance the survival likelihood of COVID-19 patients at these late stages of disease[29,30]. Therefore, antiviral compounds endowed with anti-inflammatory activity and broad tissue bioavailability would be highly desirable as ideal anti-COVID-19 drug candidates. To date, this has only been experimentally accomplished by the complex administration of dexamethasone and antibodies to SARS-CoV-2[31]. In this work, we characterized that kinetin (MB-905) inhibits SARS-CoV-2 replication in vitro, in vivo and reduces virus-induced enhancement of IL-6 and TNFα levels. As a nucleobase, MB-905 is converted into its triphosphate nucleotide to inhibit viral RNA synthesis and induce error-prone virus replication. MB-905 shows good oral absorption, its metabolites are stable, achieving long-lasting plasma and lung concentrations, and this drug is not mutagenic nor cardiotoxic in acute and chronic treatments.

## Results

### MB-905 inhibits SARS-CoV-2 replication and virus-induced enhancement of IL-6 and TNF levels

To discover potential leads against SARS-CoV-2, we tested a library of nitrogenous bases, nucleoside, and nucleotide analogs in virus-infected human hepatoma (Huh-7) cells, in a tumor cell line that recapitulates type II pneumocytes (Calu-3)[32,33] and in primary monocytes. N6-furfurylaminopurine (kinetin), designated MB-905 (Supplementary Fig. 1a), inhibited SARS-CoV-2 replication in a concentration-dependent manner in huh-7 and calu-3 cells (Supplementary Fig. 2, green line). By blocking the N9 position (MB-906, Supplementary Fig. 1b) or adding an unphysiological excess of adenine, the antiviral activity of MB-905 was impaired (Supplementary Fig. 2, black lines), suggesting that MB-905 may require activation by the host cell through the adenine phosphoribosyl transferase pathway (APRT), similar to FPV[34]. Considering that APRT could convert MB-905 to its riboside 5′-monophosphate and that 5′-nucleotidases would convert it to the corresponding nucleoside, these two potential prodrugs were synthesized, the phosphoroamidate MB-711 and the nucleoside MB-801 (Supplementary 1C, D). MB-905, MB-801, and MB-711 showed similar in vitro potencies to inhibit SARS-CoV-2 replication in calu-3 cells (Fig. 1a and Supplementary Table 1). We further demonstrated that their efficacies were improved by treating the cells twice, just postinfection and 24 h later (Fig. 1b and Supplementary Table 1). Cytotoxicity was not enhanced when treatment was performed just once or twice (Supplementary Table 1 and Supplementary Fig. 3). Although adenine-related MBs were less potent than RDV, MB-905, MB-801 and MB-711 had 6- to 40-fold higher selective indexes than molnupiravir. These adenine-derived compounds displayed one order of magnitude lower cytotoxicity than molnupiravir (Supplementary Table 1).

Next, we tested whether MB-905, its related compounds, as well as the controls RDV and molnupiravir, could reduce SARS-CoV-2 RNA levels and virus-induced inflammatory markers in human primary monocytes. In contrast with molnupiravir, which showed relatively low activity in SARS-CoV-2-infected monocytes, MB-905, MB-801, MB-711 and RDV reduced cell-associated virus RNA levels in a concentration-dependent manner (Fig. 1c). Additionally, we observed that molnupiravir was highly cytotoxic to these myeloid cells. With molnupiravir at 10 μM, monocyte viability was drastically reduced (Fig. 1c and Supplementary Fig. 3c). Of note, at 10 μM, MB-905 also diminished SARS-CoV-2-induced TNF and IL-6 levels (Fig. 1d, e), a desired feature for antivirals against COVID-19.

### MB-905 induces error-prone SARS-CoV-2 RNA synthesis

Considering that MB-905 is most likely a pro-drug of its corresponding nucleotide triphosphate, we further tested whether inhibition of virus replication could be related to a direct action on SARS-CoV-2 RNA polymerase (Fig. 2a). Kinetin riboside 5′-triphosphate inhibits viral RNA polymerase but with an $IC_{50}$ 3-fold higher than that of the active triphosphate form of RDV (GS-443902) (Fig. 2a). Additionally, by measuring cell-associated genomic and subgenomic viral RNA in SARS-CoV-2-infected Calu-3 cells, we found that MBs actively impaired viral RNA synthesis (Fig. 2b). Whereas RDV inhibited both the markers of replication (genomic RNA) and transcription (subgenomic RNA) at similar magnitudes, MBs were more effective in reducing replication than transcription (Fig. 2b). Considering that MB-905 and related prodrugs deliver riboside 5′-triphosphate, which could be a substrate for SARS-CoV-2 RNA polymerase (Fig. 2a), N6-furfuryladenine would be incorporated into the virus genome from MB-905-treated SARS-CoV-2-infected Calu-3 cells. To test this hypothesis, high affinity anti-N6-furfuryadenine IgG was used to immunoprecipitate (IP) virus RNA, followed by quantification by RT–PCR. Nil-treated cells already presented a basal level of kinetin in the viral RNA compared to control isotype immunoprecipitation (Fig. 2c), suggesting a natural occurrence of N6-furfuryladenine in the viral genome – which could be due to a natural oxidation of nucleic acid components during sample preparation. Remarkably, viral RNA obtained from anti-N6-furfuryladenine IP of MB-905-treated SARS-CoV-2-infected Calu-3 cells was enriched by more than 10-fold viral RNA levels compared to isotype control and nil-treated cells (Fig. 2c). Thus, N6-furfuryladenine seems to be incorporated into the viral genome upon treatment with MB-905. We next sequenced the full-length virus

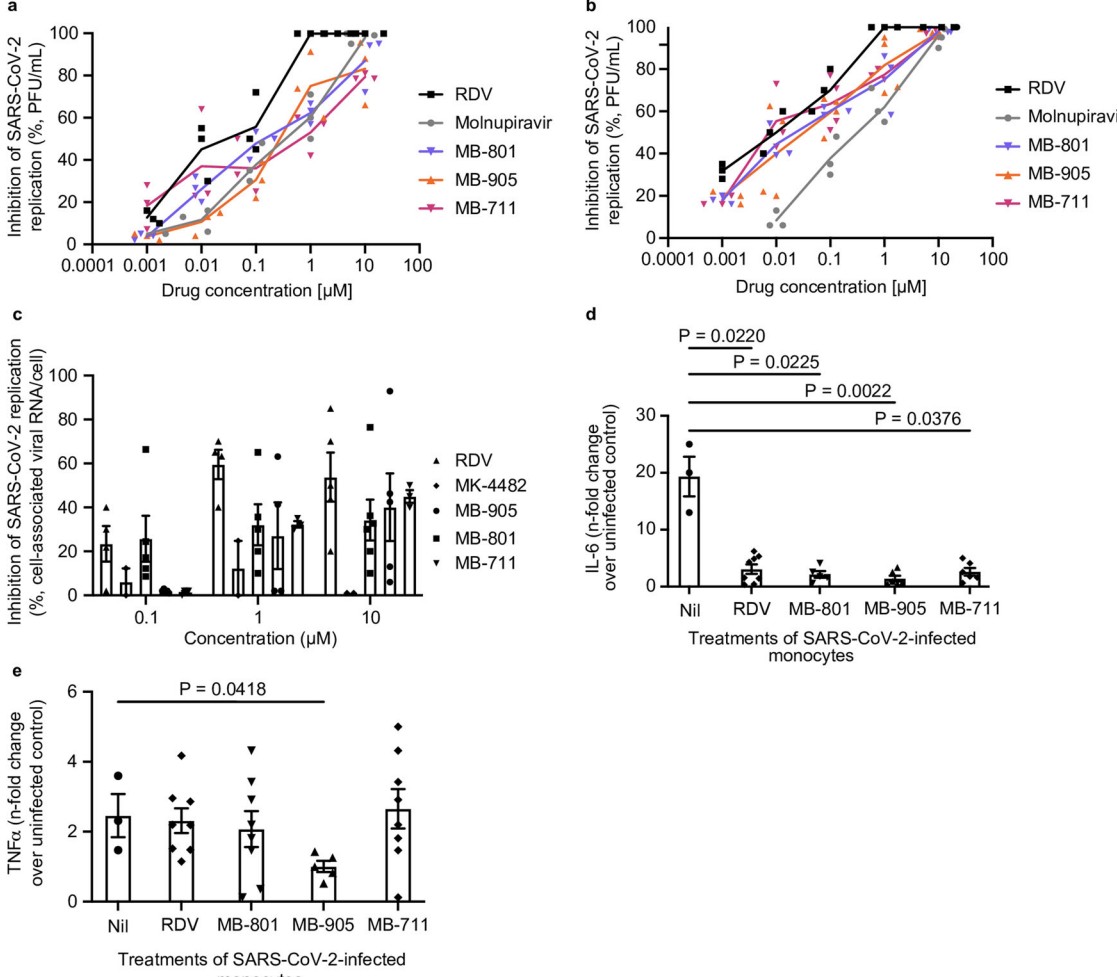

**Fig. 1 | The antiviral and anti-inflammatory activities of the MBs against SARS-CoV-2.** Calu-3 cells at a density of $5 \times 10^4$ cells/well in 96-well plates were infected with SARS-CoV-2 at an MOI of 0.1 for 1 h at 37 °C. The inoculum was removed, the cells were washed and incubated with fresh DMEM containing 2% fetal bovine serum (FBS), and the indicated concentrations of the compounds were added immediately thereafter ($n = 3$) (**a**), or the cells were also treated an additional time at 24 h postinfection ($n = 3$) (**b**). After 48 h, cell supernatants were harvested, and infectious viral titers in the culture supernatant were measured by PFU/mL in Vero cells. Human primary monocytes were infected at an MOI of 0.1 and treated with the indicated concentrations of the compounds. After 24 h, cell-associated virus RNA loads ($n = 3–6$) (**c**), as well as TNF-α ($n = 3–8$) (**d**) and IL-6 ($n = 3–8$) € levels in the culture supernatant, were measured. In panel **c**, different concentrations of the antivirals were tested, whereas in panels **d** and **e**, the drugs were used at 10 μM. The data represent the means ± SEMs of experiments with cells from at least three healthy donors of monocytes. Kruskal-Wallis one-way ANOVA test was used to determine *P* values, because donors are no matching/paring, it is hard to assume Gaussian distribution and no correction for multiple comparisons were made.

genome from MB-905- and nil-treated SARS-CoV-2-infected cells and observed increased changes at the first base level, especially T(U) → A and C → A (Fig. 2d). The error-prone replication made the virus populations grown in the presence and absence of MB-905 phylogenetically distinguishable (Fig. 2e). The cumulative effects of four SARS-CoV-2 passages in calu-3 cells in the presence of MB-905 and molnupiravir were comparable (Fig. 2f).

Modeling the presence of MB-905 in a double-stranded RNA suggests that its N6-furfuryl moiety may impact the overall structure of the nucleic acid and base pairing (Supplementary Fig. 4a). Whereas when the N6-furfuryl moiety of MB-905 is projected towards the outer area of the double-stranded RNA, it does not affect its interaction with uracil, it is also energetically favorable that the N6-furfuryl moiety interacts internally with other nucleobases (Supplementary Fig. 4a). As suggested by changes in root-mean-square deviation (RMSD) values from 0.31 to 2.1 (dimensionless scores) when N6-furfuryl is projected towards the outer and inner regions of the double-stranded RNA (Supplementary Fig. 4a), MB-905 will most likely make RNA strand distances wider. In addition, internal projection of N6-furfuryl bumps into neighboring adenine and uracil, forming non-Watson-Crick interactions with other residues, such as guanine, in a Wobble-like conformat ion[35,36] (Supplementary Fig. 4b–g). Noncanonical base pairing could lead to error-prone replication.

## MB-905 synergizes with anti-SARS-CoV-2 exonuclease repurposed drugs

Knowing that the proof-reading mechanism in the SARS-CoV-2 replication complex could actively excise the incorporated nucleotide derived from MB-905, we tested whether the coinhibition of the exonuclease enhanced the in vitro antiviral activity of MBs under investigation. Both HIV integrase[16] and HCV NS5A[37] inhibitors have been proposed to target SARS-CoV-2 exonuclease. Thus, MB-905 or control RNA polymerase inhibitors were combined with suboptimal doses of dolutegravir (DTG), raltegravir (RTG), pibrentasvir (PIB), ombitasvir (OMB), or daclatasvir (Supplementary Table 2). We observed that the combination of RNA polymerase and repurposed inhibitors of SARS-CoV-2 nsp14 enhanced the potency of the former, such as MB-905 and the control antivirals (Supplementary Table 2).

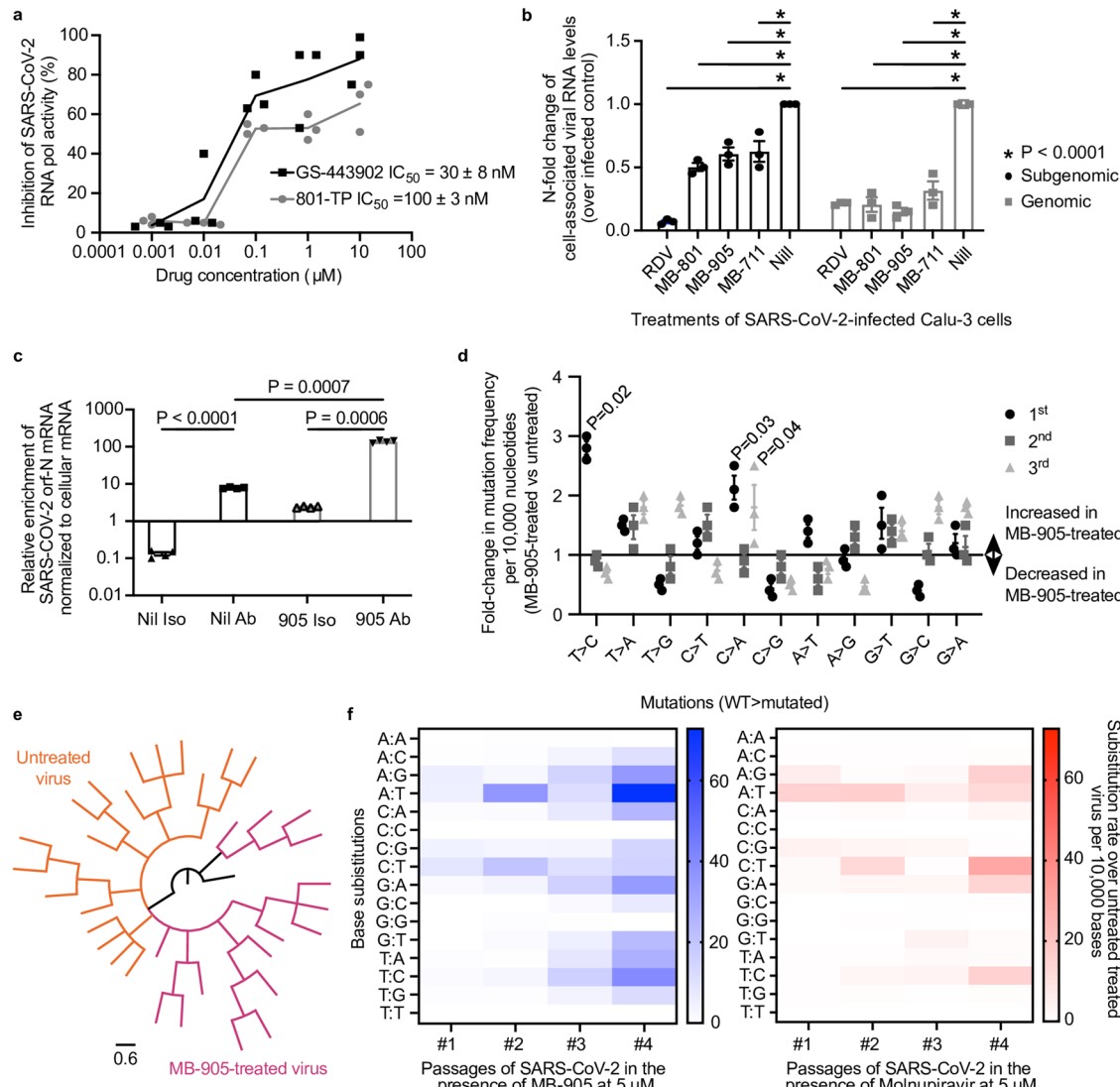

**Fig. 2 | MB905 inhibits SARS-CoV-2 RNA synthesis.** SARS-CoV-2 RNA polymerase was incubated with a template/primer, NTPs and $MgCl_2$ for 3 h at 37 °C in the absence and presence of the antiviral MB-905-ribose (801) triphosphate (801-TP) and GS-443902 ($n = 3$). Pyrophosphate release was quantified by luminescent assay (Lonza Bioscience, LT07-610). ($n = 3$) (**a**). Cell monolayers, of calu-3-infected cells treated with 10 μM of the compounds for 48 h, were lysed, total RNA was extracted,- and quantified by RT–PCR to ORF1 and ORFN mRNA ($n = 3$; mean values ± SEM; ordinary two-way ANOVA) (**b**). Cell-associated viral RNA from mock- and SARS-CoV-2-infected calu-3 cells treated or not with MB-905 was incubated with anti-kinetin antibody (Ab) or nonspecific IgG (isotype control; Iso) coupled with protein A conjugated to magnetic beads. After washing, ORFN mRNA and cellular housekeeping genes (GAPDH) were quantified by real-time RT–PCR ($n = 4$; mean values ± SEM; ordinary two-way ANOVA). SARS-CoV-2 was passaged in Huh-7 cells in the presence of increasing concentrations, 0.5 to 9 μM, of MB-905 and sequenced ($n = 3$; mean values ± SEM; ordinary one-way ANOVA). Mutation profile (**d**) and virus phylogeny (**e**) are presented. Four passages of SARS-CoV-2 in Calu-3, untreated or treated with MB-905 or molnupiravir at 5 mM ($n = 4$/group), was sequenced and compared in terms of mutations (**f**). Only sequences eith quality scores above Q30 and average coverage above 10,000-fold were used. Consensus sequences are deposited on GISAID: #EPI_ISL_402125,#EPI_ISL_1023783-EPI_ISL_1023845 and raw data on SRA: #SRR19181788-SRR19181796, SRR19181801, SRR191806-SRR19181809, SRR19181812 and SRR19181813.

In the case of MB-905, efficient inhibition at the EC99 level was achieved when combined with HIV integrase inhibitors (Supplementary Table 2). To better demonstrate the enhancement achieved for MB-905, MB-801, or MB-711 in combination with 5 μM DTG or RTG, the results are presented as virus productive titers in untreated and treated virus-infected cells (Fig. 3a, b). Whereas MBs inhibited virus replication by approximately 1−$\log_{10}$ at 10 μM, combination with 5 μM RTG (Fig. 3a) or DTG (Fig. 3b) increased antiviral inhibition to approximately 2−$\log_{10}$. Of note, RTG or DTG alone showed marginal effects on SARS-CoV-2 replication when tested at 5 μM (Fig. 3a, b). In particular, MB-905 synergized with DTG (Fig. 3c), which is a drug with very favorable pharmacokinetics[38]. These results on drug combination not only reinforce the characterization of the MB-905 mechanism of action as an error-prone molecule but also

indicate that our lead compound could be used together with repurposed drugs.

**Nonclinical safety and oral availability of MB-905**
MB-905 leads to error-prone virus replication, but in contrast with mutagenic molnupiravir[19], it was negative for both Ames and micronucleus tests (Supplementary Tables 3 and 4). Additional preclinical studies for MB-905 were performed, including pharmacokinetics in mice and rats, maximum tolerable dose, 28-day repeated oral dose, toxicokinetic assay, in vitro and in vivo potential severe cardiac side effects, and plasma albumin binding (Fig. 4a, Supplementary Figs. 5−7, and Supplementary Tables 5 and 6). When assessed in vivo in mice and rats, MB-905 displayed over 50% oral bioavailability (Fig. 4a, b, Supplementary Fig. 5a, b, and Supplementary Tables 5 and 6). MB-905 was

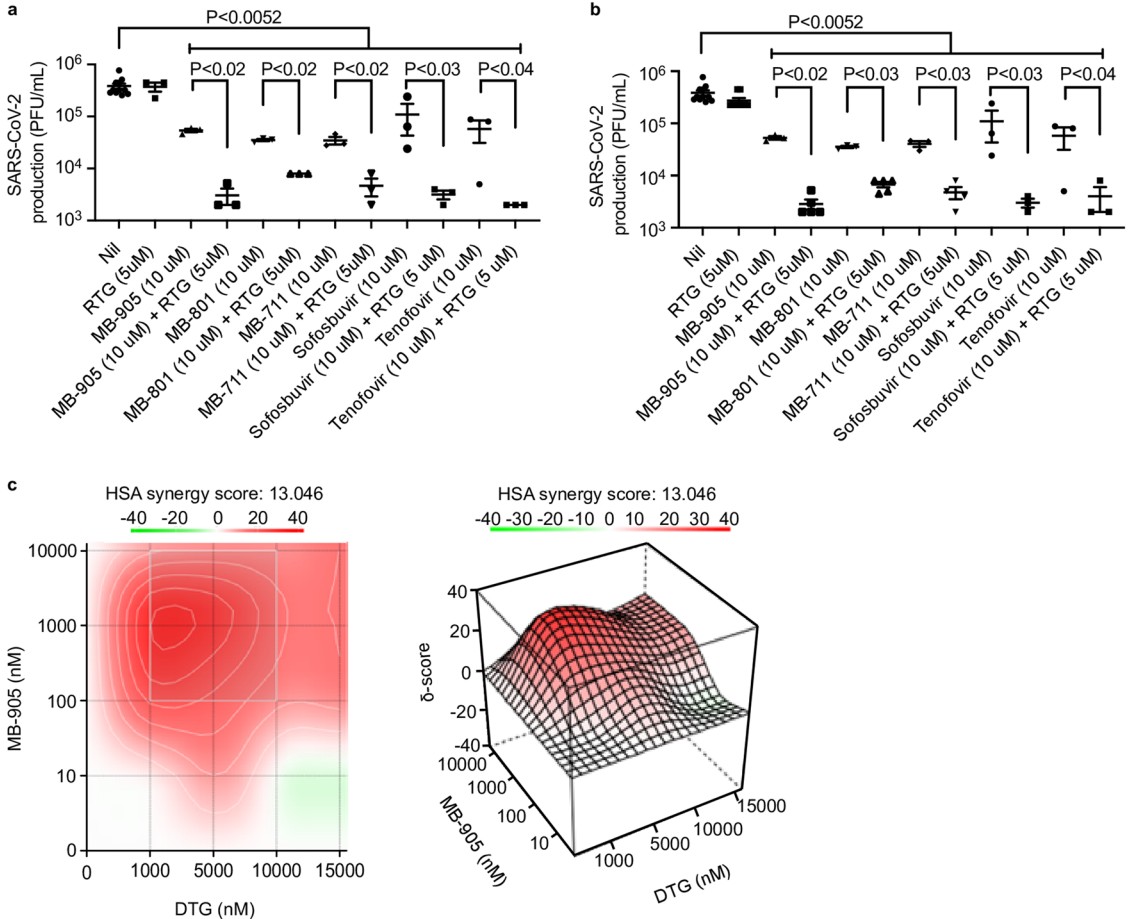

**Fig. 3 | Synergistic inhibition of SARS-CoV-2 replication through combination of MB-905 and repurposed viral exonuclease inhibitors.** Infected calu-3 cells were treated with the indicated concentrations of MBs with raltegravir (RTG) (**a**) or dolutegravir (DTG) (**b**), followed by titration in Vero cells. Ordinary one-way ANOVA test was used to compare Nil vs the other groups, $P = 0.0052$ represents the minimum statistical significance, between Nil and sofosbuvir. For specific comparisons between two groups, treated with a nucleic acid component analog vs its combination with RTG (**a**) or DTG (**b**) one-tailed Student's T test was performed. Infected calu-3 cells were treated with the full curve of MB-905 or dolutegravir, followed by titration in Vero cells and analysis for synergy using SinergyFinder2.0 (https://synergyfinder.fimm.fi/) (**c**). Scores above 10 indicate synergy. The data represent the means ± SEMs of three independent experiments.

found in plasma and lungs (Fig. 4b, c). MB-905-derived metabolites were detected (Fig. 4d and Supplementary Figs. 5 and 6), with molecular weights consistent with its nucleoside (equivalent to MB-801, metabolites 1 and 4 from Fig. 4d and Supplementary Figs. 5 and 6), nucleotide monophosphate (metabolites 2 and 3 from Fig. 4d and Supplementary Figs. 5 and 6) and triphosphate (metabolite 5 from Fig. 4c and Supplementary Figs. 5 and 6). In lung extracts, either the nucleoside, nucleotide monophosphate or triphosphate (metabolites 1, 2, and 5, respectively, from Fig. 4c) was more abundant, consistent with the notion that nucleosides undergo phosphorylation in the target tissue for COVID-19 antiviral activity. Moreover, incubation of MB-711 with mouse liver extracts, followed by measurement of 5′-nucleotidase activity, revealed that this molecule could be used as a substrate in the substitution of AMP (Supplementary Fig. 8). We interpret that upon APRT activation, MB-905 riboside monophosphate may be converted to its corresponding nucleoside (MB-801) to be released in the plasma. Altogether, these data suggest that oral administration of MB-905 leads to the formation of its active metabolite in the respiratory tract to act as a potent inhibitor of the polymerase and thus inhibit virus reproduction.

**SARS-CoV-2-infected transgenic mice expressing human ACE2 and hamsters are protected by MB-905**

For a subsequent evaluation of the in vivo activity of MB-905 on SARS-CoV-2-infected mice, we aimed to reach the concentration that could be later translated into clinical trials. Because MB-905's active principal

is kinetin, which has been used in patients with familial dysautonomia[39,40] at doses of 11.1 mg/kg/day or above, reaching plasma exposure within the anti-SARS-CoV-2 pharmacological parameters, infected mice were treated with an equivalent dose. According to standard dose conversion between species[41], 140 mg/kg/day in mice would be equivalent to 11.1 mg/kg/day in humans.

Indeed, MB-905 exposure in the plasma (Fig. 5a) and lungs (Fig. 5b) of uninfected mice treated with 140 and 500 mg/kg (MB-905's NOAEL) were above the in vitro potency against SARS-CoV-2 in calu-3 cells. MB-905's metabolites, including its nucleoside, nucleotide-mono- and nucleotide-triphosphate, lasted in the lungs of uninfected mice for nearly 24 h (Fig. 5c). Transgenic mice expressing human ACE2 (K18-hACE2) infected with a lethal challenge of SARS-CoV-2 ($10^5$ PFU of gamma VoC) and treated with 140 mg/kg/day (since 12 h postinfection) survived after two weeks (Fig. 5d) and recovered their body weight (Fig. 5e).

To explore more conditions, the next experiments were performed with K18-hACE2 over a one-week time frame. SARS-CoV-2-infected mice were treated with MB-905 at 140 mg/kg/day or its half-dose, 70 mg/kg/day, combined or not with DTG at 10 mg/kg/day. Consistent with the in vitro synergy results, the HIV integrase inhibitor enhanced the survival of MB-905-treated mice upon infection (Fig. 6a). Under treatment with MB-905 at 140 mg/kg/day alone and in combination with DTG, infected mice sustained their body masses beyond the threshold of euthanasia (Fig. 6b) and improved their overall clinical

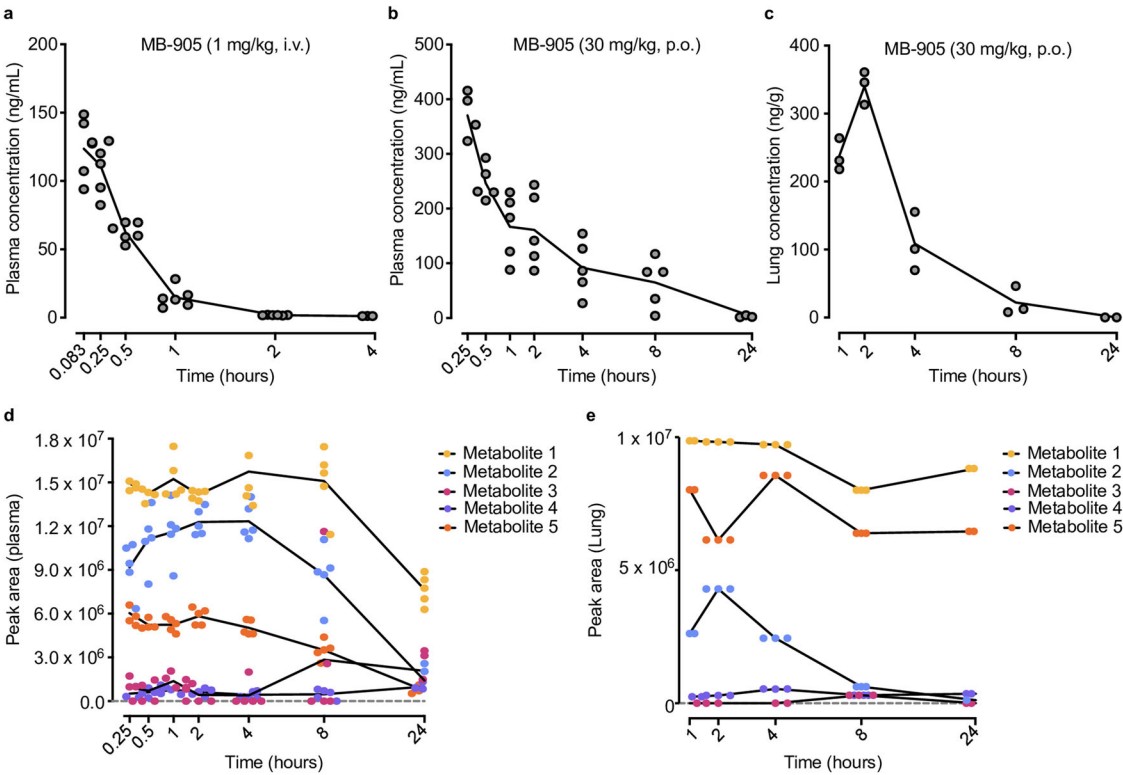

**Fig. 4 | Pharmacokinetic and putative metabolites of MB-905 in rats. a** Single intravenous dose (1 mg/kg bodyweight) pharmacokinetics properties of MB-905 in rat plasma ($n = 6$); **b** single oral dose (30 mg/kg bodyweight) pharmacokinetics properties of MB-905 in rat plasma ($n = 5$); **c** single oral dose (30 mg/kg bodyweight) pharmacokinetics properties of MB-905 in lung ($n = 5$); **d** plasma metabolites (5 metabolites) after treatment with MB-905 (30 mg/kg, p.o.) ($n = 5$); **e** lung metabolites (5 metabolites) after treatment with MB-905 (30 mg/kg, p.o.). Data are expressed as the mean ± SEM (standard error of the mean). Non-compartmental data analysis was performed using Phoenix WinNonlin®.

scores (Fig. 6c). Although MB-905 administered with or without DTG did not alter virus RNA levels from lung extracts (Fig. 6d), the infectious titers decreased significantly upon treatment (Fig. 6e).

Alone or in combination with DTG, MB-905 reduced lung necrosis because the levels of the intracellular marker LDH were decreased in bronchoalveolar lavage (BAL) upon treatment (Fig. 6f). Furthermore, treatments reduced pulmonary insults provoked by SARS-CoV-2 (Fig. 6g). Whereas infected/untreated mouse lung histology displayed collapsed alveoli septum and intense hemorrhage, treated animals displayed a lung parenchyma closer to mock-infected mice (Fig. 6h, purple for alveoli and red for erythrocytes). Reduced damage in the lungs of infected mice treated with our drug or combinations was associated with decreased levels of double-stranded RNA (dsRNA), a biomarker of SARS-CoV-2 replication, in the lungs (Fig. 6i, amber-colored cells for dsRNA). SARS-CoV-2-induced inflammation in vivo was attenuated in the BAL fluid of MB-905-treated mice (Supplementary Fig. 9).

In contrast to K18-hACE2 mice, which, once infected, rapidly unfolds to a lethal outcome, hamsters facilitate observations of antiviral interventions. The dosage to be used was based on our most complete preclinical safety profile based on toxicological experiments on animals larger than mice. The experimental rats in our study tolerated MB-905 well at 250 mg/kg (Supplementary Fig. 7). According to standard conversion calculations between species[41], 250 mg/kg in rats is equivalent to 296 mg/kg in hamsters. With this background, we performed a twice daily treatment in hamsters. A regime closer to potential future clinical investigations. For practical reasons, of drug dilution and combinations, we aimed to use 140 mg/kg twice a day (BID), which results in a daily exposure below the equivalent dose of preclinical support. Both mock- and SARS-CoV-2-infected hamsters tolerated the doses of 140 mg/kg once a day (QD) and BID well (Fig. 7a).

Similar to infected mice, MB-905 BID reduced infectious virus titers in the lungs (Fig. 7b) and, consequently, protected this organ in the hamsters (Fig. 7c–e). The effect of MB-905 on reducing viral load and protecting against lung injury was reproduced in different animal models.

## Discussion

Currently, the only direct-acting treatment against SARS-CoV-2 recommended by the WHO is Paxlovid[23]. For this protease inhibitor, resistance mutations have been proposed[7]. Thus, new antivirals with a different mechanism of action are necessary to fight COVID-19. We found that MB-905, an adenine analog with furfuryl radical at the N-6 position, impaired SARS-CoV-2 RNA synthesis, induced error-prone replication, with increased U → C and C → A changes, and synergized with viral exonuclease inhibitors. MB-905 was safe, well tolerated, orally available, and decreased the pulmonary damage provoked by SARS-CoV-2 in K18-hACE2-mice and hamsters.

Based on viral sequencing and in silico studies, we interpret that MB-905 can act as tautomeric purine, in which C6-N could either act as a donor or acceptor for hydrogen bonds, depending on whether hydrogen at position C6-N interacts with oxygen from the furfuryl group. The codon bias observed for MB-905 is supported by other independent studies on a rare disease called familial dysautonomia, in which kinetin has been proposed as a treatment. While healthy individuals have the base T at the 6th nucleotide position of the 20th exon of the gene that encodes the protein kinase complex associated with I-κ-B (IKBKAP), patients with familial dysautonomia have base C at this nucleotide position[42,43]. This T → C change affects the splicing process of the intron adjacent to that exon[42,43]. Kinetin allows correct splicing to occur, suggesting that it is incorporated into the intronic RNA present as an A/G analog, which are purines able to act as donors and

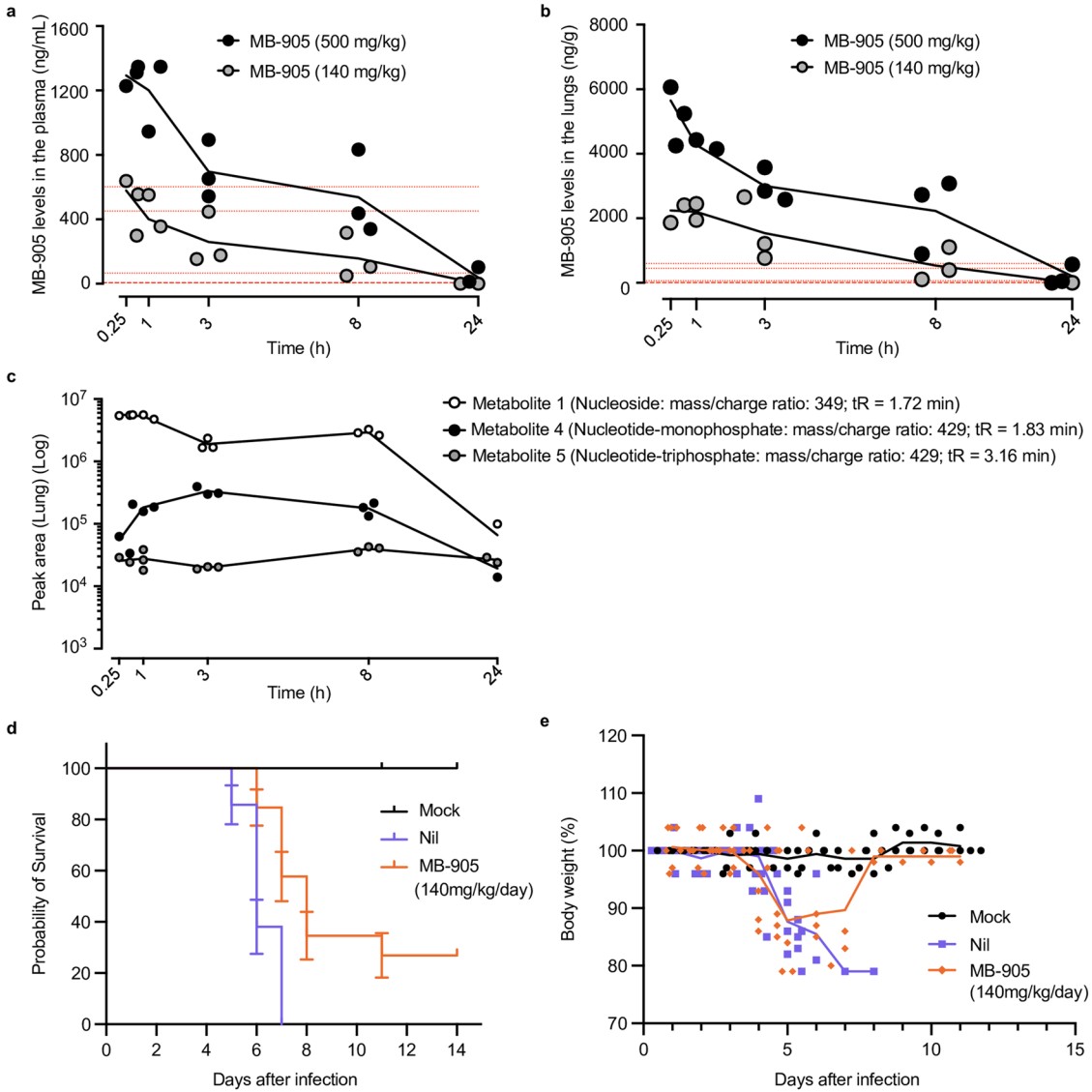

**Fig. 5 | Pharmacokinetic and pharmacodynamics in mice.** Single oral treatment of MB-905 in mouse plasma ($n = 6$) (**a**) and lungs ($n = 6$) (**b**), dotted red lines represent the values of $EC_{50}$ and $EC_{90}$ for MB-905 when added once or twice to calu-3-infected cells. Lung metabolites were detected consistently in terms of peak area and duration (**c**). Transgenic mice expressing the hACE2 receptor to SARS-CoV-2 entry at the age of 10–12 weeks were infected with $10^5$ PFU intranasally and treated orally beginning 12–18 h after infection with 140 mg/kg/day. Survival curves [$n = 13$ (mock), 21 (nil), 26 (MB-905)] (**d**) and percentage of mouse body weight [$n = 5$ (mock), 9 (nil), 20 (MB-905)] (**e**) are presented.

acceptors of hydrogen bonds, respectively, at position 6 of the nucleobase. Therefore, there is a convergence between the antiviral mechanism and the effect of kinetin to overcome innate genetic errors. Additionally, kinetin has been used in humans as an anti-aging, anti-apoptotic and anti-inflammatory drug[44–47]. In line with these studies, MB-905 attenuated SARS-CoV-2-induced inflammation.

Previous phase I/II clinical trials carried out in patients with familial dysautonomia showed that kinetin doses between 11.1 and 23.5 mg/kg/day were well tolerated[39,40]. In these trials, kinetin reached human plasma exposures from 8 to 23 μM[40], which are above our in vitro $EC_{90}$ values for MB-905, in the range of 2.1–6.3 μM. Thus, considering this range of $EC_{90}$ values for MB-905 and according to the results on kinetin pharmacokinetics in humans[40], plasma exposures would last for 8 to 12 h above the threshold of virus inhibition. The in vivo dosages of 140 mg/kg QD in mice and BID in hamsters were shown to be protective to the host organism and are, according to standard conversion calculations between species[41], equivalent in humans to be in the range of 11.38–18.92 mg/kg. Therefore, our in vitro and in vivo results are consistent with the doses already used in humans, and the detection of the

active ribosyl-triphosphate metabolite in the lungs of treated mice confirms the suggested activation mechanism.

Our preclinical investigation indicates that MB-905 may be clinically developed not only as monotherapy but also in combination with repurposed drugs that inhibit SARS-CoV-2 nsp14/10, such as HIV integrase inhibitors[16]. Among this class of compounds, dolutegravir has the most favorable pharmacokinetics[38]. When administered at the dose of 50 mg per day in humans, dolutegravir achieves plasma concentrations between 15 and 1.5 μM at the points of $C_{max}$ and $C_{min}$, respectively[38]. These concentrations synergized in vitro with MB-905. MB-905 in vitro pharmacological parameters were improved to 5–10 times when combined with dolutegravir at 5 μM. Altogether, our results could motivate clinical trials with MB-905 at doses of 11.5 mg/kg, twice daily, combined or not with standard dolutegravir posology for patients with COVID-19. The proposed dose and regimen of MB-905 against COVID-19 would be lower and shorter than the 23.5 mg/kg/day used for 28 days in humans in previous studies[39,40]. Furthermore, in a more recent clinical trial, kinetin was used orally at a dosage of 30 mg/kg/day by patients with familial dysautonomia for 3 years

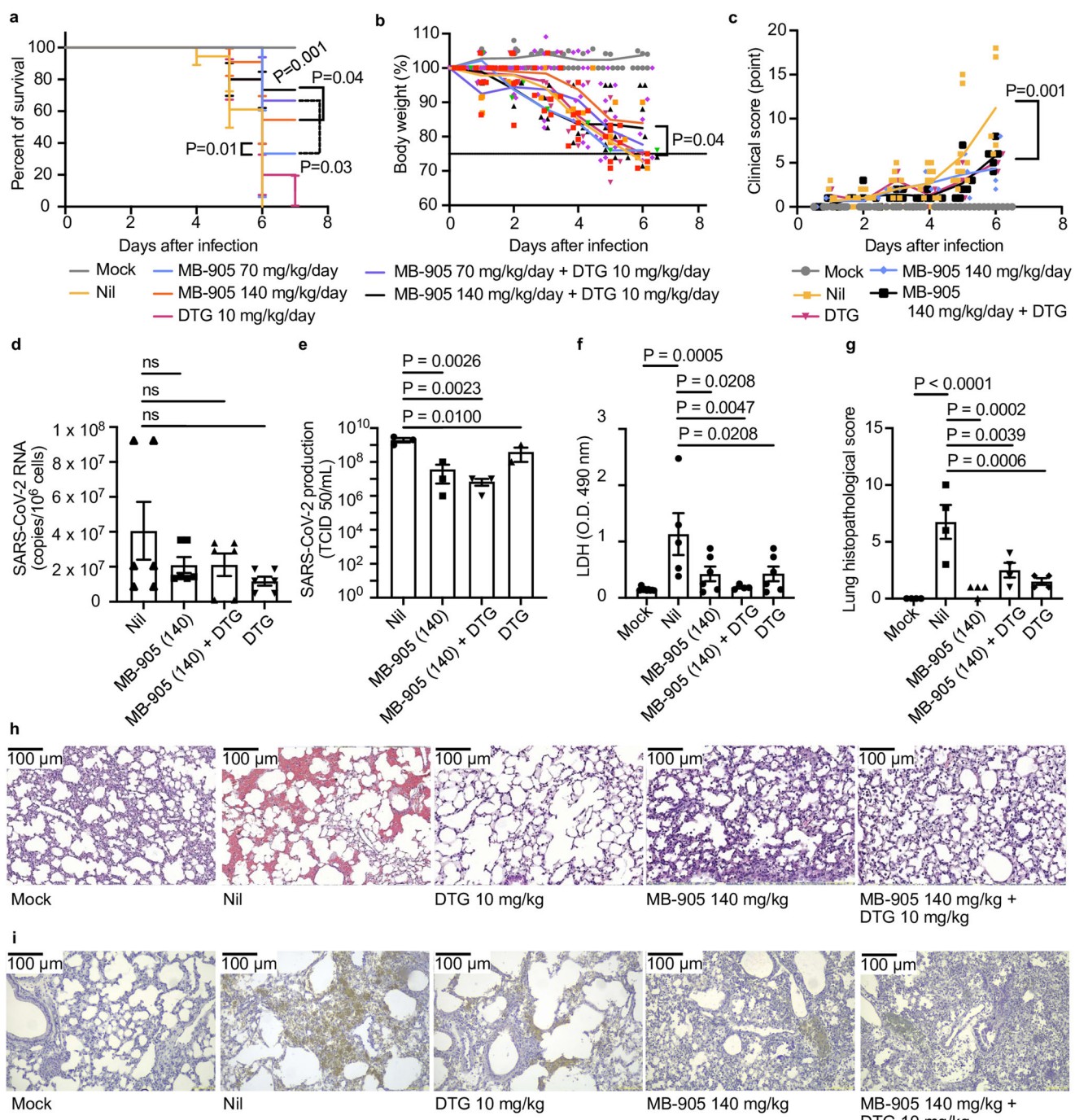

**Fig. 6 | MB-905 antiviral activity in transgenic K18 mice infected with SARS-CoV-2 gamma VoC.** Transgenic mice expressing the hACE2 receptor to SARS-CoV-2 entry at the age of 10–12 weeks were infected with $10^5$ PFU intranasally. After 12–18 h, the treatments were performed and maintained daily. **a** 7-day survival summary of infected and treated animals ($n = 14$ mock, 18 nil, 9 MB-905 70 mg/kg/day, 11 MB-905 140 mg/kg/day, 10 DTG, 8 MB-905 70 mg/kg/day + DTG, 15 MB-905 140 mg/kg/day +DTG). **b** Change in body weight ($n = 8$ mock, 8 nil, 8 MB-905 70 mg/kg/day, 11 MB-905 140 mg/kg/day, 8 DTG, 8 MB-905 70 mg/kg/day + DTG, 15 MB-905 140 mg/kg/day +DTG) were tested by simple linear regression. **c** Clinical score variation ($n = 8$ mock, 9 nil, 3 DTG, 3 MB-905 140 mg/kg/day, 6 MB-905 140 mg/kg/

day +DTG) was tested by simple linear regression. The groups treated with MB905 at 140 mg/kg/day, alone or in combination with dolutegravir (DTG), and their controls were analyzed for viral RNA levels ($n = 6$ animals/group) (**d**) and titers ($n = 3$) (**e**) in the lungs and LDH activity in the BAL ($n = 4$–9 animals/group) (**f**). Histological scores ($n = 4$) (**g**), representative images of hematoxylin-eosin staining (**h**, purple for alveoli and red for erythrocytes) and immunohistochemistry for dsRNA (**i**, amber-colored cells for dsRNA) from three independent experiments are presented. Data from panels **d**–**g** represent the mean ± SEM tested by ordinary one-way ANOVA. Scale bar represents 100 μm.

without interruption[48], thus reinforcing that MB-905 is a highly safe and promising candidate against COVID-19. Naturally, to treat a human genetic disease, kinetin would have to be administered chronically, while its potential use against COVID-19 would be limited to a much narrower period.

In summary, our results reconfirm the paramount importance of nucleoside analogs as a rich source of lead molecules capable of inhibiting the replication of biomedically important viruses[49]. We demonstrated that MB-905, a molecule with easy synthetic access and cost effectiveness, is a prodrug of the corresponding nucleoside

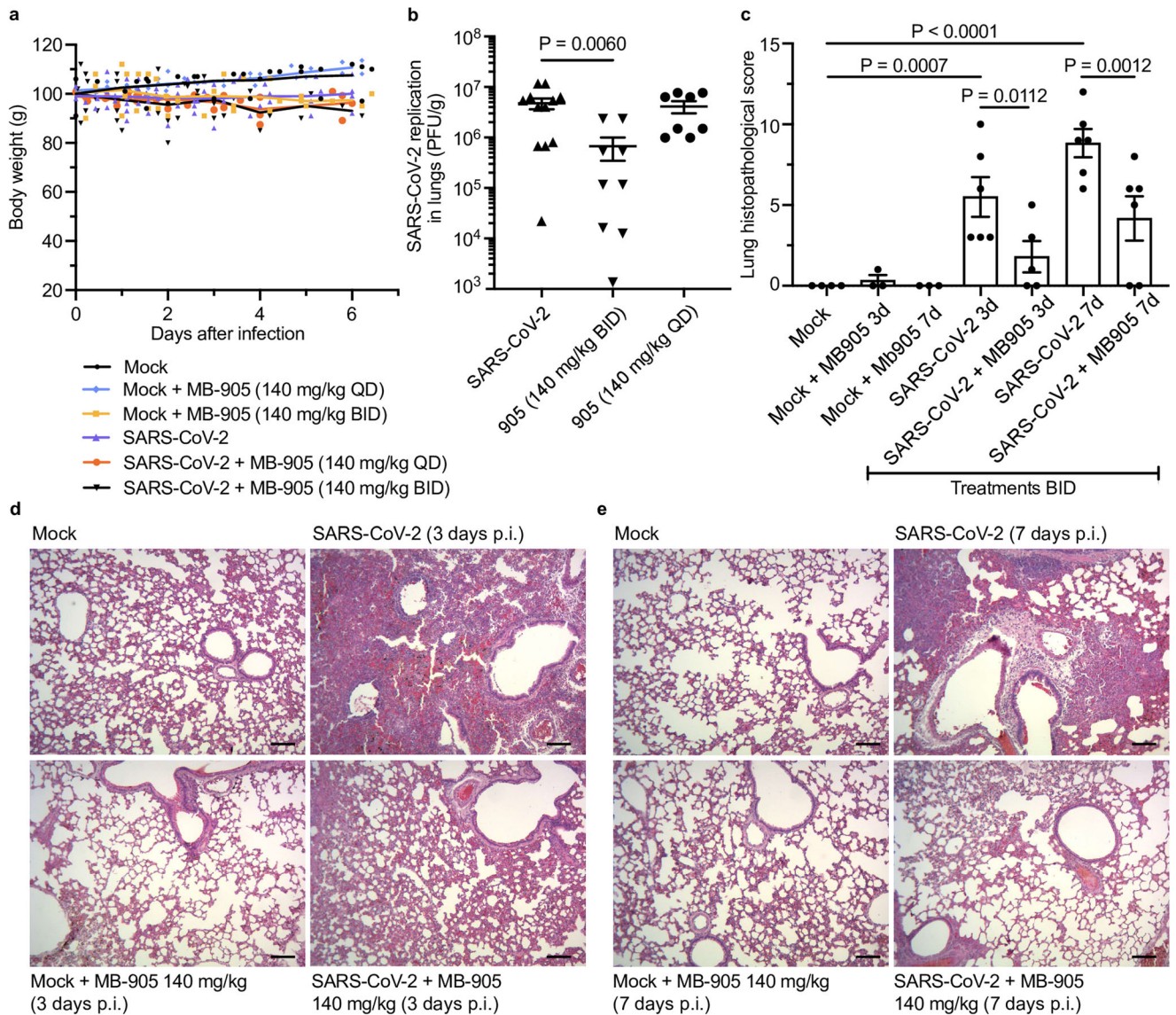

**Fig. 7 | MB-905 reduces SARS-CoV-2-induced lung injury in hamsters.** Male and female 6- to 8-week-old Syrian hamsters were inoculated (intranasally) with 100 μL of saline (mock) or SARS-CoV-2 ($10^5$ PFU). Animals were analyzed for changes in body weight ($n = 4$ Mock, 6 Mock + MB905 140 mg/kg BID, 7 Mock + MB905 140 mg/kg QD, 12 nil/SARS-CoV-2, 8 SARS-CoV-2 + MB905 140 mg/kg BID and 8 SARS-CoV-2 + MB-905 140 mg/kg QD) (**a**). Infectious virus titers were determined at the 3rd day after infection ($n = 12$ nil/SARS-CoV-2, 9 SARS-CoV-2 + MB905 140 mg/kg BID and 8 SARS-CoV-2 + MB-905 140 mg/kg QD) (**b**). Histological scores in the

lungs were determined at the 3rd and 7th day after infection ($n = 4$ Mock, 3 Mock + MB905 140 mg/kg at the 3rd day, 3 Mock + MB905 140 mg/kg at the 7th day, 6 nil/ SARS-CoV-2 at the 3rd day, 5 SARS-CoV-2 + MB905 140 mg/kg at the 3rd day, 6 nil/ SARS-CoV-2 at the 7th day and 6 SARS-CoV-2 + MB-905 140 mg/kg at the 7th day) (**c**). Data are presented as mean ± SEM and were tested by ordinary one-way ANOVA. Representative H&E staining of hamster lungs from the 3rd (**d**) and 7th (**e**) days after infection of three independent experiments are presented. Scale bar represents 100 μm.

triphosphate that inhibits SARS-CoV-2 RNA synthesis and therefore its replication in relevant models of COVID-19 pathophysiology. MB-905 affected the viral pattern of nucleotide recognition, and this activity may be potentiated by inhibiting the viral 3′−5′-exonuclease. Oral administration of MB-905 reduced SARS-CoV-2-induced lung damage in vivo. Our preclinical development of MB-905 as a new antiviral compound together with its previous clinical investigation in familial dysautonomia strongly indicates the promising opportunity to advancing MB-905 into clinical trials against COVID-19[9,14,50]. This possibility is under consideration in Brazil.

## Methods
This research complies with ethical regulations. Experiments of infection of K18-hACE2- mice were performed in the Animal Biosafety Level 3 (ABSL-3) multiuser facility, according to the animal welfare

guidelines of the Ethics Committee of Animal Experimentation from National Cancer Institute José Alencar Gomes da Silva (CEUA-INCa, Licence 005/2021). Hamster infection was carried out in an ABSL-3 facility at Instituto de Ciências Biomédicas from the Federal University of Minas Gerais (UFMG), under Ethical approval by the Committee for Animal Experimentation of the UFMG (process no. 165/2021). Preclinical experiments in CD-1 mice, Swiss mice and Sprague-Dawley rats were performed in accordance to the Ethics Committee approvals (CEUA # 210, 214, 215, 217, 241, and 306) of the Center of Innovation and Preclinical Studies (CIEnP). The use of primary human cell was approved by Institutional Review Board from the Oswaldo Cruz Foundation under protocol 49971421.8.0000.5248 with signed informed consent in accordance with the Hemotherapy Service of Hospital Clementino Fraga from the Federal University of Rio de Janeiro. All volunteers were de-identified.

## Reagents

The antivirals RDV, tenofovir and molnupiravir were purchased from Selleckhem (https://www.selleckchem.com/). Sofosbuvir and the MBs (Supplementary Fig. 1) were synthetized by Microbiologica Química-Farmacêutica LTDA (Rio de Janeiro, Brazil). ELISAs were purchased from R&D Bioscience. Recombinant SARS-CoV-2 proteins were purchased from BPS Biosciences (https://bpsbioscience.com/). For in vitro experiments, all small molecule inhibitors were dissolved in 100% dimethylsulfoxide (DMSO) and subsequently diluted at least $10^4$-fold in culture or reaction medium before each assay. The final DMSO concentrations showed no cytotoxicity. The materials for cell culture were purchased from Thermo Scientific Life Sciences (Grand Island, NY), unless otherwise mentioned.

**Small molecule synthesis.** All solvents were used as received from commercial suppliers. All of the other reagents used in the synthesis were purchased from Sigma–Aldrich except the reagent for the phosphoramidation reaction ((S)-isopropyl 2-(((S)-(perfluorophenoxy) (phenoxy) phosphoryl) amino) propanoate), purchased from ALCHIMIA LIMITED. All reactions were carried out under an argon atmosphere. Reactions were monitored with analytical TLC on silica gel 60-F254 precoated aluminum plates and visualized under UV (254 nm). Purification was performed by column chromatography on silica gel (35–70 μM) or by crystallization. NMR data were recorded on a Bruker Fourier 80, or Bruker 500 MHz spectrometer in the deuterated solvents indicated, and the spectra were calibrated on TMS. Chemical shifts (δ) are quoted in ppm, and $J$ values are quoted in Hz. In reporting spectral data, the following abbreviations were used: s (singlet), d (doublet), t (triplet), q (quartet), dd (doublet of doublets), dddd (doublet of double doublets) and m (multiplet). Mass spectrometry analysis was performed on a Shimadzu LCMS-8045. HPLC was carried out on a Shimadzu Prominence chromatograph equipped with a DAD detector (Xterra RP18, 250 × 4.6 mm 5 mm – part n°:186000496 column). Solvents were used as HPLC grade. Structural details are highlighted in the Supplementary Chemical information.

**Synthesis of Kinetin, MB-905 (A).** In a three-neck round-bottom flask equipped with a reflux condenser, a thermometer, an addition funnel and mechanical stirring, a suspension of 6-chloropurine (50 g, 0.323 mol) in ethanol (500 mL) was cooled to 5 °C. A solution of furfurylamine (57 mL, 0.647 mol) and 4-dimethylaminopyridine (0,4 g, 3.2 mmol) in ethanol (50 mL) was slowly added. Then, the system was heated under reflux for 4 hours. After this time, the system was cooled to 5 °C, and the crystalline solid was filtered and washed with 150 mL of a 1:1 ethanol/water solution. The crude product was recrystallized from ethanol. The resulting white crystalline solid was washed with cold ethanol and dried under vacuum. Yield: 66 g, 95%, m.p. 265–272 °C dec, mass spectra LC–MS (m/z): [M]$^+$ = 216.10, $^1$H NMR (500 MHz, DMSO-d6) δ 12.99 (s, 1H), 8.20 (s, 1H), 8.11 (s, 2H), 7.54 (s, 1H), 6.35 (s, 1H), 6.22 (s, 1H), 4.69 (s, 2H) ppm, $^{13}$C NMR (126 MHz, DMSO-d6) δ 154.0, 153.2, 152.3, 149.7, 141.8, 139.1, 119.0, 110.5, 106.6, 36.5 ppm, HPLC (X Terra RP18, 250 × 4.6 × 5 mm, isocratic 10% fase B [MeOH: MeCN (1:1)]: 90% fase A [3,4 g/L KH$_2$PO$_4$ pH 2,54 (H$_3$PO$_4$ 4 4 1.0 mL/min, 210 nm, 30 °C) $t_r$ = 8.768 min, λmax 272 nm.

**Synthesis of 6-furfurylamino-9-(tetrahydropyran-2-yl)−9H-purine, MB-906 (B).** To a suspension of kinetin (10 g, 0.046 mol) in ethyl acetate (150 mL), 3,4-dihydropyrane (8.5 mL, 0.092 mol) was added followed by the addition of 8 mL of formic acid. The mixture was refluxed for 5 hours. After this time, TLC analysis confirmed the total consumption of the starting material. The reaction mixture was cooled and basified with a saturated aqueous solution of NaHCO$_3$. The organic phase was washed with water (2 × 50 ml), dried with anhydrous sodium sulfate, and evaporated to a yellow residue that was crystallized from ethanol (85 ml). The resulting white crystalline solid was washed with

cold ethanol and dried under vacuum. Yield: 12 g, 86%, m.p. 138–140 °C, mass spectra LC–MS (m/z): [M]$^+$ = 300.15, $^1$H NMR (80 MHz, DMSO-d$_6$) δ (ppm): 8.43 (s, 1H), 7.93 (s, 1H), 7.55–7.19 (m, 2H), 6.27 (d, $J$ = 1.4 Hz, 2H), 5.70 (dd, $J$ = 7.5, 4.3 Hz, 1H), 4.90 (d, $J$ = 5.8 Hz, 2H), 4.15 (dd, $J$ = 11.2, 2.8 Hz, 1H), 3.98–3.39 (m, 1H), 2.14–1.50 (m, 6H). $^{13}$C NMR (20 MHz, DMSO-d$_6$) δ (ppm): 154.55, 153.11, 152.00, 148.75, 141.95, 137.72, 119.61, 110.33, 107.29, 81.67, 68.64, 37.63, 31.74, 24.84, 22.78, HPLC (X Terra RP18, 250 × 4.6 × 5 mm, gradient 5–100% phase B [90% ACN/10% H2O (0.1% TFA)]: phase A [90% H2O (0.1% TFA)/10% ACN], 1.2 mL/min, 210 nm, 30 °C) $t_r$ = 14.443 min, λmax 264 nm.

**Synthesis of Kinetin riboside, MB-801 (D).** To a suspension of 6-chloropurine riboside (10 g, 0,034 mol) in EtOH (100 mL), furfurylamine (6 mL, 0.069 mmol) was added followed by dropwise addition of Et$_3$N (15 mL, 0.102 mol). The system was refluxed for 5 h. After this time, TLC analysis confirmed the total consumption of the starting material. The product was filtered. The crude product was recrystallized from EtOH, filtered and washed with cold EtOH. After drying in vacuo, the product was obtained as a white crystalline solid. Yield: 8.7 g (90% yield), m.p. 152–154 °C, mass spectra LC–MS (m/z): [M]$^+$ = 348.15, [α]$_D^{25}$ −60.7 (c 1.0, EtOH), $^1$H NMR (80 MHz, DMSO-d$_6$) δ (ppm): 8.39 (s, 1H), 8.27 (d, $J$ = 4.4 Hz, 2H), 7.54 (s, 1H), 6.50–6.30 (m, 1H), 6.23 (d, $J$ = 3.2 Hz, 1H), 5.91 (d, $J$ = 6.0 Hz, 1H), 5.46 (d, $J$ = 6.1 Hz, 1H), 5.33–5.03 (m, 1H), 4.94–4.48 (m, 3H), 4.17 (s, 1H), 3.98 (dd, $J$ = 3.4 Hz, 1H), 3.76–3.52 (m, 2H). $^{13}$C NMR (20 MHz, DMSO-d$_6$) δ (ppm): 154.59, 152.98, 152.48, 148.97, 142.03, 140.28, 119.92, 110.65, 106.94, 88.28, 86.16, 73.81, 70.90, 61.89, 36.41, HPLC (X Terra RP18, 250 × 4.6 × 5 mm, gradient 5–100% phase B [90% ACN/10% H2O (0.1% TFA)]: phase A [90% H2O (0.1% TFA)/10% ACN], 1.2 mL/min, 210 nm, 30 °C) $t_r$ = 14.443 min, λmax 264 nm.

**Phenyl (Isopropoxy-L-alaninyl) Kinetin Riboside Phosphoramidate, MB-711 (C).** A flame-dried round-bottom flask flushed with argon was charged with kinetin riboside 2′,3′ acetonide (prepared according to the literature procedure[51]) (500 mg, 1.29 mmol). The flask was flushed again with argon, and anhydrous THF (10 mL) was added by syringe. The solution was cooled to 0–5 °C in an ice bath, and t-BuMgCl (2 M in THF, 0.8 mL, 1.55 mmol) was slowly added by syringe. Next, the mixture was stirred at 0–5 °C for 40 min. After that time, neat reagent was added for phosphoramidation. The mixture was stirred for 12 h at room temperature. AcOH (1 M, 5 mL) was then added to quench the reaction before the solvent was removed under reduced pressure to leave the crude product as a pale-yellow oil, which was used in the next step without further purification. Next, a solution of the crude product in DCM:ethylene glycol 1:1 (10 mL) was treated with 30 mol% camphorsulfonic acid (89 mg, 0.38 mmol) at 5 °C, and the mixture was stirred for 16 h at this temperature. After that time, TLC analysis confirmed the total consumption of the starting material. The solvent was removed under reduced pressure, and the crude product was purified by flash column chromatography and then recrystallized from ethanol. The resulting light yellowish crystalline solid was washed with cold ethanol and dried under vacuum. Yield: 575 mg, 68% (after two steps), m.p. 138–140 °C, mass spectra LC–MS (m/z): [M]$^+$ = 216.10 = 617.30, [α]$_D^{25}$ −22.5 (c 0.2, CH$_2$Cl$_2$), $^1$H NMR (500 MHz, CDCl$_3$) δ 8.31 (s, 1H), 7.99 (s, 1H), 7.33 (s, 1H), 7.17 (t, $J$ = 7.8 Hz, 2H), 7.09 (d, $J$ = 8.1 Hz, 2H), 7.01 (t, $J$ = 7.3 Hz, 1H), 6.65 (s, 1H), 6.28 (s, 2H), 5.96 (d, $J$ = 3.2 Hz, 1H), 4.92 (hept, $J$ = 6.2 Hz, 1H), 4.79 (sl, 2H), 4.42 (d, $J$ = 3.0 Hz, 2H), 4.32 (dddd, $J$ = 31.6, 14.0, 8.7, 3.1 Hz, 4H), 3.88 (ddt, $J$ = 16.3, 9.2, 7.1 Hz, 1H), 3.70 (q, $J$ = 7.0 Hz, 1H), 1.28 (d, $J$ = 7.1 Hz, 3H), 1.22 (t, $J$ = 7.0 Hz, 1H), 1.16 (t, $J$ = 6.2 Hz, 6H) ppm. $^{13}$C NMR (DEPT-135) (126 MHz, CDCl$_3$) δ 173.2, 173.2, 154.4, 152.8, 151.6, 150.5, 149.7, 142.4, 138.6, 129.7, 125.1, 120.1, 120.0, 110.5, 107.7, 89.7, 83.7, 83.5, 75.2, 70.8, 69.5, 66.2, 58.4, 50.4, 37.6, 21.7, 21.7, 21,0, 20.9, 18.5 ppm, HPLC (X Terra RP18, 250 × 4.6 × 5 mm, gradient 5–100% phase B [90% MeCN/10% H$_2$O

(0.1% TFA)]: phase A [90% H$_2$O (0.1% TFA)/10% MeCN], 1.2 mL/min, 210 nm, 30 °C) $t_r$ = 14.443 min, λmax 264 nm.

## Cells and virus

The African green monkey kidney (Vero, subtype E6), human hepatoma (Huh-7) and human lung epithelial cell line (Calu-3) were cultured in this study. Vero and Calu-3 cells were maintained in high-glucose DMEM, whereas Huh-7 cells were maintained in low-glucose DMEM. DMEM was complemented with 10% fetal bovine serum (FBS; HyClone, Logan, Utah), 100 U/mL penicillin and 100 μg/mL streptomycin (Pen/Strep; Thermo Fisher) at 37 °C in a humidified atmosphere with 5% CO$_2$.

Human primary monocytes were obtained after 3 h of plastic adherence of peripheral blood mononuclear cells (PBMCs). The use of primary human cell was approved by Institutional Review Board from the Oswaldo Cruz Foundation under protocol 49971421.8.0000.5248 with signed informed consent in accordance with the Hemotherapy Service of Hospital Clementino Fraga from the Federal University of Rio de Janeiro. All volunteers were de-identified. PBMCs were isolated from healthy donors by density gradient centrifugation (Ficoll-Paque, GE Healthcare). PBMCs (2.0 × 10$^6$ cells) were plated onto 48-well plates (NalgeNunc) in RPMI-1640 without serum for 2–4 h. Nonadherent cells were removed, and the remaining monocytes were maintained in DMEM with 5% human serum (HS; Millipore) and penicillin/streptomycin. The purity of human monocytes was above 95%, as determined by flow cytometric analysis (FACScan; Becton Dickinson) using anti-CD3 (BD Biosciences, cat # 555342, 1:40), anti-CD16 (Southern Biotech, cat # 561313, 1:20) and APC mouse IgG2a isotype (BD Biosciences, cat # 550882, 1/40) and PE mouse IgG1 isotype (BD Biosciences, 555749, 1/20).

The SARS-CoV-2 B.1 lineage (https://www.ncbi.nlm.nih.gov/nuccore/MT710714) and gamma variant (also known as P1 lineage; #EPI_ISL_1060902, hCoV-19/Brazil/AM-L70-71-CD1739/2020) were isolated on Vero E6 cells from nasopharyngeal swabs of confirmed cases. All procedures related to virus culture were handled at a biosafety level 3 (BSL3) multiuser facility according to WHO guidelines. Virus titers were determined as plaque forming units (PFU)/mL. Virus stocks were kept in −80 °C ultralow freezers.

## Cytotoxicity assay

Monolayers of 1.5 × 10$^4$ cells in 96-well plates were treated for 3 days with various concentrations (semilog dilutions from 1000 to 10 μM) of the antiviral drugs. Then, 5 mg/ml 2,3-bis-(2-methoxy-4-nitro-5-sulfophenyl)−2H-tetrazolium-5-carboxanilide (XTT) in DMEM was added to the cells in the presence of 0.01% N-methyl dibenzopyrazine methyl sulfate (PMS). After incubating for 4 h at 37 °C, the plates were measured in a spectrophotometer at 492 and 620 nm. The 50% cytotoxic concentration (CC$_{50}$) was calculated by a nonlinear regression analysis of the dose–response curves.

## Yield-reduction assay

Vero E6 cells were infected with a multiplicity of infection (MOI) of 0.01. HuH-7 cells, Calu-3 cells, and monocytes were infected at an MOI of 0.1. Cells were infected at densities of 5 × 10$^5$ cells/well in 48-well plates for 1 h at 37 °C. The cells were washed, and various concentrations of compounds were added to DMEM with 2% FBS. After 24 h (Vero E6 cells), 48 h (Calu-3 cells), or 72 h (HuH-7 cells) supernatants were collected, and harvested virus was quantified by PFU/mL or real-time RT−PCR. A variable slope nonlinear regression analysis of the dose–response curves was performed to calculate the concentration at which each drug inhibited the virus production by 50% or 90% (EC$_{50}$ and EC$_{90}$, respectively).

## Virus titration

Monolayers of Vero E6 cells (2 × 10$^4$ cell/well) in 96-well plates were infected with serial dilutions of supernatants containing SARS-CoV-2 for 1 h at 37 °C. Fresh semisolid medium containing 2.4% carboxymethylcellulose (CMC) was added, and the culture was maintained for 72 h at 37 °C. Cells were fixed with 10% formaldehyde for 2 h at room temperature and then stained with crystal violet (0.4%). The virus titers were determined by plaque-forming units (PFU) per milliliter.

Cells were washed, fresh medium supplemented with 2% FBS was added, and 3 to 5 days post infection, the cytopathic effect was scored in at least 3 replicates per dilution by independent readers. The reader was blinded to the source of the supernatant.

## Molecular detection of viral RNA levels

Total viral RNA from a culture supernatant and/or monolayers was extracted using QIAamp Viral RNA (Qiagen®) according to the manufacturer's instructions. Quantitative RT–PCR was performed using GoTaq® Probe qPCR and RT-qPCR Systems (Promega) in a StepOne™ Real-Time PCR System (Thermo Fisher Scientific) ABI PRISM 7500 Sequence Detection System (Applied Biosystems). Amplifications were carried out in 25 μL reaction mixtures containing 2× reaction mix buffer, 50 μM of each primer, 10 μM of probe, and 5 μL of RNA template. Primers, probes, and cycling conditions recommended by the Centers for Disease Control and Prevention (CDC) protocol were used to detect SARS-CoV-2[52]. The standard curve method was employed for virus quantification. For reference to the cell amounts used, the housekeeping gene RNAse P (F-5′-AGATTTGGACCTGCGAGCG-3′, R-5′GAGCGGCTGTCTCCACAAGT-3′, P-5′-FAM-TTCTGACCTGAAGGCTCT GCGCG-BHQ-1-3′) was amplified. The Ct values for this target were compared to those obtained for different cell amounts, 10$^7$ to 10$^2$, for calibration. Alternatively, genomic (ORF1; F-5′-ATGAGCTTAGTCCTG TTG-3′, R-5′-CTCCCTTTGTTGTGTTGT-3′, P−5′-FAM-AGATGTCTTGTG CTGCCGGTA-BHQ-1-3′) and subgenomic (ORFE; F-5′ACAGGTAC GTTAATAGTTAATAGCGT-3′, R-5′-ATATTGCAGCAGTACGCACACA-3′, P-5′-FAM-ACACTAGCCATCCTTACTGCGCTTCG-BHQ-1-3′) were detected, as described elsewhere[53].

## Generation of mutant virus

Calu-3 cells were infected with SARS-CoV-2 at an MOI of 0.1 for 1 h at 37 °C and then treated with MB905 at increasing concentrations, from 0.5 to 9 μM, after each passage. That is, after infection and initial treatment, cells were accompanied daily up to the observation of cytopathic effects (CPE), and virus was recovered from the culture supernatant, titered and used in the next round of infection in the presence of a higher drug concentration of the drug. As a control, SARS-CoV-2 was also passaged in the absence of treatments to monitor genetic drift associated with culture passage. Virus RNA virus was extracted by Qiamp viral RNA (Qiagen) and quantified using a Qubit 3 Fluorometer (Thermo Fisher Scientific) according to the manufacturer's recommendations.

RNA viruses were submitted to unbiased sequencing using an MGI-2000 and a metatranscriptomics approach. To do so, at least 4.2 ng of purified total RNA from each sample was used for library construction using the MGIEasy RNA Library Prep Set (MGI, Shenzhen, China). All libraries were constructed through RNA-fragmentation (250 bp), followed by reverse-transcription and second-strand synthesis. After purification with MGIEasy DNA Clean Beads (MGI, Shenzhen, China), the samples were submitted to end-repair, adaptor-ligation, and PCR amplification steps. After purification as previously described, samples were quantified with a Qubit 1X dsDNA HS Assay Kit using an Invitrogen Qubit 4.0 Fluorometer (Thermo Fisher Scientific, Foster City, CA) and homogeneously pooled (1 pmol/pool of PCR products) and submitted to denaturation and circularization steps to be transformed into a single-stranded circular DNA library. Purified libraries were quantified with a Qubit ssDNA Assay Kit using Invitrogen Qubit 4.0 Fluorometer (Thermo Fisher Scientific, Foster City, CA), and DNA nanoballs were generated by rolling circle amplification of a pool (40

fmol/reaction), quantified as described for the libraries, loaded onto the flow cell and sequenced with PE100 (100-bp paired-end reads) or PE250 (100 bp paired-end reads).

Sequencing data were initially analyzed in the usegalaxy.org platform. In Brief, FASTQ raw genomic data were demultiplexed for the analysis of double-ended amplicon data by preprocessed the files in the FASTP v.0.20.1 software to remove adapters and reads smaller than 50 bp (-l 50). Reference mapping and genome assembly were performed with BWA-MEM v. 0.7.17, using Wuhan-hu-1 genome (GISAID: #EPI_ISL_402125) as the reference. The output BAM files were subsequently quality filtered (-q 30) and reformatted with SAMTools view v.1.13 to exclude (-F) unmapped reads. BAM files were trimmed for primers, using BED file containing the initial coordinates of MGI's ATOPlex v2.0 panel, by the platform iVar trim v.1.3.1 (-m 1 -q 0 -s 4 -e) and realigned to the reference genome with LoFreq v.2.1.5, adding qualities indel based on Dindel Algorithm. Variants were called with iVar variants v.1.3.1 (-q 30 -t 0.51 –pass_only) and VCF output files were used to call consensus with bcftools v.1.10. SARS-CoV-2 consensus genomes have been aligned and assigned to global outbreak lineages with Pangolin v.3.1.17. Mutations were assigned according to Snpeff Version 5.1. Alignment was performed through ClustalW using Mega 7.0 software. This software was also used to construct the evolutionary history of the sequencing passages by maximum likelihood method with the Kimura-2 parameter model, with 1000 bootstraps. The phylogenetic tree was rooted by the Wuhan-01 index case.

## In silico studies

The 3D structure corresponding to the genetic sequence of SARS-CoV-2 at position 1681 (GAUCGCCAUU, upon RNA-dependent RNA polymerase reaction)[54] was built at pH 7.4, and the structure was energy minimized by density functional theory (DFT) under the dielectric constant of water by Spartan'18 software (Wavefunction, Inc., Irvine, CA, USA; https://www.wavefun.com/). For the structure with the lowest energy, the base adenine was replaced by a kinetin moiety, and a novel energy minimization under the same conditions described above was conducted. The superposition between the genetic sequence of SARS-CoV-2 (template) and each replacement condition corresponding to the three confirmations with lowest energy positions was generated with PyMOL Delano Scientific LLC software (DeLano Scientific LLC; https://pymol.org/2/) with the statistical approximations through the commands align, cealign and super and was considered the approximation with the lowest root mean square deviation (RMSD) value. This same software was also used to generate the figures and detect the main interactions among kinetin and nucleobases through a cut-off for the interaction of 4 Å[55] and analysis of van der Waals radius superposition.

## RNA immunoprecipitation

Calu-3 cells ($2 \times 10^6$) were infected with SARS-CoV-2 at an MOI of 0.1 and treated with 10 μM of the compounds. After 24 h, the supernatant was removed, and the monolayer was washed with PBS. Monolayer RNA was extracted by a commercial kit (Qiagen) following the manufacturer's recommendation. To the extracted RNA, we added 25 μL of anti-kinetin/protein A/magnetic beads (1:1:1). Anti-kinetin polyclonal antibody was purchased from Agrisera LTD, Sweden (www.grisera.com; cat #AS09444; 1:1000), and nonspecific rabbit sera were used as an isotype control (BD Biosciences, cat# 550875, 1:25). After overnight shaking at 4 °C, magnetic racks allowed us to wash the materials with 175 μL of TBS-T (Tris Buffer Solution 1 M, MgCl$_2$ 1 M and NaCl 5 M) buffer. After that, RNA was re-extracted using a viral RNA mini kit (Qiagen), and samples were subjected to RT–PCR to detect ORFN (F- 5′GACCCC AAAAT-CAGCGAAAT-3′, R-5′-TCTGGTTACTGCCAGTTGAATCTG-3′, P-5′-FAM-ACCCCGCATTACGTTTGGTGGACC-BHQ-1-3′) mRNA and a housekeeping gene (GAPDH; ThermoFischer, cat # Hs02786624_g1). One-step real-time RT–PCR conditions are described above.

## SARS-CoV-2 RNA polymerase inhibition assay

The SARS-CoV-2 RNA polymerase, nsp12, 7, and 8 (BPSBiosciences # 100839), was purchased and incubated under assay conditions described elsewhere[14], with modifications in the assay readout. In brief, 500 nM of the 33-1-mer template and 10-mer primer described by Wang et al.[14] and synthesized by IDT (www.idtdna.com) was incubated with 125 nM of the viral RNA polymerase in a volume of 20 μL containing 250 μM of each nucleotide triphosphate (NTP), 50 mM HEPES (pH 7.0), 50 mM NaCl, 5 mM MgCl$_2$, 4 mM DTT and the antiviral nucleotide analogs for 3 h at 25 °C. Kinetin ribose 5′-triphosphate (MB-801TP) and GS-443902 (equivalent to remdesivir triphosphate) were purchased (https://www.biolog.de/6-fu-atp; https://www.medchemexpress.com/gs-443902.html). The reaction mixture was quenched with an equal volume of 20 mM EDTA (pH 8.0). Next, serial dilutions of the stopped reaction were incubated with a pyrophosphate luminescent detection kit (PPIlight # LT07-610; https://bioscience.lonza.com/lonza_bs/BR/en/). Reaction luminescence was quantified in a Glomax Discover plate reader (www.promega.com).

## 5′-Nucleotidase assay

Theis enzymatic activity was measured by a commercial assay #ab235945 from www.abcam.com. The commercial substrate AMP or MB-711 was incubated with mouse liver extracts potterized in 10% sucrose from euthanized mock-infected mice. Liver extracts were a source of 5′-nucletidase, and in the case of MB-711, the enzymes cathepsin A or carboxylesterase 1 and histidine triad nucleotide-binding protein 1 (HINT1) release its nucleotide monophosphate.

## Preclinical development of MB-905

All in vivo procedures were performed in accordance with the Ethical Principles in Animal Experimentation adopted by the National Council for the Control of Animal Experimentation (CONCEA). The protocol used in all in vivo studies reported here was reviewed and approved by the Ethics Committee on Animal Use (CEUA # 210, 214, 215, 217, 241, and 306) of the Center of Innovation and Preclinical Studies (CIEnP). The mice and rats used in the following assays were obtained from the CIEnP bioterium, whose breeding colonies were purchased from Charles River Laboratories (USA) and were maintained under specific pathogen-free (SPF) conditions. All experiments included Male and female animals.

## Pharmacokinetics in rodent plasma and lung tissue

Pharmacokinetics in rodent plasma was performed in CD1 mice (*Mus musculus*, 20–30 g) and Sprague–Dawley rats (*Rattus norvegicus*, 250–300 g) of both sexes. The following preformulations were used to dissolve MB-905: a dose of 3 mg/kg (i.v.): 1% DMSO + 4% PEG 400 + 0.5% Tween 80 + 94.5% saline; dose of 30 mg/kg (p.o.): 10% DMSO + 40% PEG 400 + 5% Tween 80 + 45% saline; dose of 550 mg/kg (p.o.): 5% Tween 80 + 95% PEG 400. This trial consisted of administering MB-905 at doses of 10, 30, or 550 mg/kg, orally or 3 mg/kg intravenously. After oral or intravenous administration, blood samples were collected at times of 0.25, 0.5, 1, 2, 4, 8, and 24 and 0.083, 0.25, 0.5, 1, 2, and 4 h. Parameters such as area under the curve (AUC), time taken to reach the maximum concentration ($T_{max}$), time taken for $C_{max}$ to drop in half ($T_{1/2}$), volume of distribution, clearance, elimination constant and bioavailability (F) were evaluated. Bioavailability was calculated using the following equation: F (%) = [(intravenous dose x oral AUC)/(oral dose × intravenous AUC)] × 100. To assess the putative metabolites of MB-905 in the lungs, Sprague−Dawley rats (8–12 weeks − 250–300 g) of both sexes were used. Animals were treated orally with MB-905 as previously described. Animals were sacrificed at different time points after treatments, and lungs were immediately removed, placed in liquid nitrogen at −80 °C and processed for further analysis. For the analysis of plasma and lung supernatant, UPLC−MS/MS equipment was used.

## Plasma protein binding

The assay consisted of the incubation of 200 μL blood plasma fractions from male and female mice containing MB-905 (0.26, 2.6, or 26 μM) and PBS using equilibrium dialysis (RED-Thermo Fisher®) with 8 kDa molecular weight cut-off membranes. After 4 h of incubation at 37 °C and $1.5 \times g$ (orbital shaker), samples were matrix-matched and quenched by protein precipitation, followed by LC–MS/MS analysis.

## Metabolic stability of compounds in HLM and rCYP inhibition

Substrate stock solutions were prepared in DMSO and diluted in methanol and PBS for a final organic composition of 0.2% DMSO and 0.5% methanol. MB-905 (0.1, 1, or 10 μM) was incubated in HLM (0.2 mg/mL) diluted in PBS (100 mM, pH 7.4) in the absence or presence of NADPH (0.6 mM) in a final volume of 100 μL. Incubations were conducted at 37 °C in triplicate. At various time points (0, 5, 15, 30, and 60 min), a 15 μL aliquot was removed and quenched with 100 μL of methanol containing the internal standard minoxidil (50 ng/mL). The rCYP inhibition assay was performed using the standard substrates midazolam and dextromethorphan for rCYPA4 and rCYP2D6, respectively. Samples were vortexed and centrifuged ($3,5123 \times g$, 5 min, 4 °C), and the supernatant used for measuring the metabolites 4-OH-midazolam and O-demetyl-dextromethorphan by UPLC–MS/MS. Data represent the comparison of metabolite production in the presence of MB-905 (100 μM) versus the presence of standard enzyme inhibitors, ketoconazole (0.5 μM, for CYP3A4) and quinidine (0.2 μM, for CYP2D6).

## Maximum tolerated dose toxicity study (MTD) and dose selection in mice

For the MTD assay (OECD 425), mice (3 animals of each sex/group) were divided into five experimental groups, and the up and down procedure was applied to which 175 mg/kg was used as the first dose. Then, since the MTD assay pointed out 550 mg/kg as the recommended dose for repeated exposure, tolerability of MB-905 was assessed by submitting two experimental groups (5 of each sex/group) to an oral treatment with the vehicle (group 1) or with MB-905 (550 mg/kg) for 7 consecutive days. Mortality, morbidity, body weight, food consumption, and general and detailed clinical signs of toxicity were evaluated. At necropsy, vital (brain, heart, liver, spleen, kidneys and adrenal glands) and reproductive organs (ovaries, testicles, and epididymis) were carefully removed, weighed and stored for further analysis (if necessary).

## Twenty-eight-day repeated-dose oral toxicity in mice

Male and female CD1 mice (6–8 weeks, 10 mice/group/sex) were treated orally with vehicle (45% polyethylene glycol 400 - PEG 400, 30% propylene glycol, 20% filtered water and 5% ethanol) or with different doses of MB-905 (10 mg/kg, 80 mg/kg or 250 mg/kg) once daily for 28 days (main animals). Additionally, recovery groups (5 males and 5 females) were established to which the same treatment regimen was applied, but animals (Vehicle or MB-905 250 mg/kg) remained untreated for another 14 days to observe persistence, reversibility, or delayed occurrence of toxic effects related to the administration of the Test Item. These experiments were conducted in compliance with the GLP principles. Morbidity, mortality, body weight, food consumption, and general and detailed clinical signs were evaluated. In the last week, urine samples from all animals were collected for analysis. At necropsy, blood samples were collected for hematological, biochemical and coagulation analyses, followed by tissue and organ collection for macroscopic and histopathological analyses.

## Prolonged toxicokinetics in mice

Mice (5 of each sex/group) were treated orally with MB-905 at doses of 10, 80, or 250 mg/kg for 28 consecutive days. Blood samples on the 0th and 28th days after treatments were collected at 0.25, 1, 3, 8, and 24 h and analyzed by UPLC–MSMS. Plasma AUC$_{all}$ and $C_{max}$ for both time points periods were calculated.

## Cardiovascular safety pharmacology in rats by telemetry

Conscious and freely moving male Sprague–Dawley rats (9–12 weeks), previously submitted to surgery for placement of the DSI™ PhysioTel hardware system implant in the abdominal aorta, were treated orally with Vehicle (5 ml/kg) or MB-905 (50 or 250 mg/kg) once a day for 7 consecutive days. Cardiovascular parameters such as systolic blood pressure, diastolic blood pressure, heart rate, and electrocardiogram were evaluated before treatments (baseline) and at 0.5, 1, 2, 3, 4, 5, 6, 7, 12, and 24 h after the treatments on days 1 and 7.

## Inhibition of voltage-dependent potassium channels of the hERG type (human ether-a-go-go related)

The HEK293 cell line ($4 \times 10e5$ cell) (BPS Bioscience, San Diego, CA, USA) expressing recombinant human ERG potassium channel (ether-a-gogo-related gene, Kv11.1) was used. The channel activity was determined using a FLIPR potassium assay kit (Molecular Devices - San Jose, CA, USA). Cells were cultivated in microplates and incubated with loading buffer for one hour at room temperature in the dark. Then, MB-905 (0.01–300 μM) or Dofetilide (0.0001–1 μM, used as positive control drug) was added to the wells and incubated for thirty minutes at room temperature in the dark. After that, the microplate was transferred to FlexStation 3 (Molecular Devices - San Jose, CA, USA) with the addition of 1 mM thallium + 10 mM potassium using automated pipetting. Data analysis was performed using SoftMax Pro Software (Molecular Devices - San Jose, CA, USA) and GraphPad Prism. The results were expressed as the percent inhibition of the hERG channel, and the mean inhibitory concentration (IC$_{50}$) was determined.

## Ames test

This test was performed using the bacterial reverse mutation assay in accordance with OECD guideline 471 and conducted in compliance with GLP principles. The preliminary assay with the Salmonella typhimurium TA 100 strain, with or without metabolic activation (S9), was carried out with MB-905 at concentrations of 8, 40, 200, 1000, and 5000 μg/plate. As cytotoxicity and mutagenicity were not observed in the absence and presence of S9, a definitive test with the strains TA 97a, TA 98, TA 100, TA 102, and TA 1535, with or without S9, was performed with the aforementioned concentrations.

## Mouse bone marrow micronucleus test

This assay was performed in mouse bone marrow in accordance with OECD guideline 474 and conducted in compliance with GLP principles. Male and female Swiss mice (*Mus musculus*, 5–10 weeks) were divided into 5 experimental groups and were treated orally with vehicle (5% Tween + 95% PEG E 400), three different doses of MB-905 (32, 125, or 500 mg/kg) for three consecutive days or with cyclophosphamide (25 mg/kg, i.p.) for 2 consecutive days. Bone marrow cells were collected and processed according to a methodology described by Schmid (1975). The ratio of polychromatic to normochromatic erythrocytes and the count of micronuclei were determined. This assay was conducted in compliance with the GLP principles.

## Mouse infections and treatment

Experiments with male and female transgenic mice expressing human ACE-2 receptor (K18-hACE2- mice, *Mus musculus*) were performed in the Animal Biosafety Level 3 (ABSL-3) multiuser facility, according to the animal welfare guidelines of the Ethics Committee of Animal Experimentation (CEUA-INCa, Licence 005/2021) and WHO guidelines. The animals were obtained from the Oswaldo Cruz Foundation breeding colony and maintained with free access to food and water at 29–30 °C under a controlled 12 h light/dark cycle and humidity of 50–58%. Experiments were performed during the light phase of the cycle.

For infection procedures, mice were anaesthetized with 60 mg/kg ketamine and 4 mg/kg xylazine and inoculated intranasally with DMEM high glucose (Mock-infected) or $10^5$ PFU of SARS-CoV-2 gamma strain in 10 μl of DMEM high glucose. A total of 18 mice per experimental group were used in three independent experiments. The animals were monitored daily for seven days for survival and body-weight analysis. In the case of weight loss higher than 25%, euthanasia was performed to alleviate animal suffering. On the last day, the bronchoalveolar lavage (BAL) from both lungs was harvested by washing the lungs once with 1 mL of cold PBS. After centrifugation of BAL (300.*g* for 5 min), the pellet was used for total and differential leukocyte counts (diluted in Turk's 2% acetic acid fluid) using a Neubauer chamber. Lactate dehydrogenase (LDH) quantification was performed with centrifuged BAL supernatant to evaluate cell death (CytoTox96, Promega, USA). Differential cell counts were performed by cytospin (Cytospin3; centrifugation at 350×*g* for 5 min at room temperature) and stained by the May-Grünwald-Giemsa method.

After BAL harvesting, the lungs were perfused with 20 mL of saline solution to remove the circulating blood. Lungs were then collected, pottered, and homogenized in 500 μL of a phosphatase and protease inhibitor cocktail Complete, mini EDTA-free Roche Applied Science (Mannheim, Germany) for 30 s using an Ultra-Turrax Disperser T-10 basic IKA (Guangzhou, China) for virus titration, RNA and cytokine quantification. Alternatively, lungs were preserved for histology.

## Hamster infection and treatment

Male and female 6- to 8-week-old Syrian hamsters (*Mesocricetus auratus*) were bred at the Department of Parasitology, ICB, UFMG. Animals received anesthesia (ketamine 200 mg/kg and xylazine 10 mg/kg, intraperitoneally) to be intranasally inoculated with 100 μL of saline or SARS-CoV-2 ($10^5$ PFU, Gamma VoC). They were housed in ventilated cages placed in an ABSL-3 facility (ICB, UFMG) at 24 °C ± 2 °C on a 12 h light/12 h dark cycle, with water and food provided *ad libitum*. Signs of disease, including ruffled fur, back arching, weight loss, and lack of activity, were monitored daily. A subset group of animals was euthanized at specific days after infection, whereas others were kept for 10 days for body weight analysis. Just before euthanasia, the hamsters were anesthetized, and blood was collected through the abdominal vena cava to measure cell counts and harvest the plasma. The lungs were then collected and snap-frozen in liquid nitrogen or fixed in 10% neutral buffered formalin. Formalin-fixed and/or frozen fragments of other target tissues (brain, liver, spleen, heart, and meninge) were also collected for additional analyses. This project received the approval of the Ethical Committee for Animal Experimentation of the UFMG (process no. 165/2021).

## Histology and Immunohistochemistry

Histological features related to the injury caused by SARS-CoV-2 infection were analyzed in the lungs of K18-hACE2 mice and hamsters. The collected material was fixed with formaldehyde (4%), dehydrated, and embedded in paraffin to obtain tissue slices through the use of a microtome. The slices were fixed and stained with hematoxylin and eosin (H&E) for microphotograph analysis. Fixed and paraffin-embedded lungs were sectioned (5 μm thickness), stained and examined microscopically. The tissue morphological alterations observed in the lungs were determined using an inflammatory score system: (i) airway inflammation (up to 4 points), (ii) vascular inflammation (up to 4 points), (iii) parenchyma inflammation (up to 5 points), and general neutrophil infiltration (up to 5 points)[56]. The staining condition for each antibody was adjusted according to our laboratory's experience.

Briefly, for immunohistochemistry, each lung section was deparaffinized and rehydrated, and antigen retrieval was achieved. To eliminate nonspecific staining, the slides were incubated with appropriate preimmune serum for 30 min at room temperature. After incubation with a primary antibody to anti-dsRNA IgG2a (Jena

Biosciences, Germany, J2, cat # RNT-SCI-10010200, 1:500)) at room temperature for 1:30 h, slides were rinsed with phosphate-buffered saline (PBS), incubated with a labeled polymer-HRP (Peroxidase AffiniPure Goat Anti-Mouse IgG (H + L) - Jackson ImmunoResearch Inc, cat # AB_10015289, 1:2000), added according to the manufacturer's instructions and incubated for 30 min. The color reaction was developed by using 3,3'-diaminobenzidine tetrachloride (DAB) chromogen solution. All slides were counterstained with hematoxylin.

## Statistical analysis

The assays were performed blinded by one professional, codified, and then read by another professional. All experiments were carried out at least three independent times, including a minimum of two technical replicates in each assay. The dose–response curves used to calculate $EC_{50}$, $EC_{90}$, and $CC_{50}$ values were generated by variable slope plots from Prism GraphPad software 9.0. The equations to fit the best curve were generated based on $R^2$ values ≥ 0.9. Student's T test, Two-way of one-way ANOVA were used to access statistically significant $P$ values <0.05. The statistical analyses specific to each software program used in the bioinformatics analysis are described above.

## Reporting summary

Further information on research design is available in the Nature Portfolio Reporting Summary linked to this article.

## Data availability

Replicates are presented in the graphics and source data files are available as online supporting materials. The source data files are provided supplementary files. The consensus sequencing data generated in this study have been deposited in the https://gisaid.org/ database under accession code: #EPI_ISL_1023783-EPI_ISL_1023845. The raw sequencing data on have been deposited in the bioproject PRJNA823058 (Severe acute respiratory syndrome coronavirus 2 (ID 823058) - BioProject - NCBI (nih.gov)) and under SRA's (https://www.ncbi.nlm.nih.gov/sra) accession codes #SRR19181788-SRR19181796, SRR19181801, SRR191806-SRR19181809, SRR19181812 and SRR19181813. Source data are provided with this paper.

## Code availability

The consensus sequencing data generated in this study have been deposited in the https://gisaid.org/ database under accession code: #EPI_ISL_1023783-EPI_ISL_1023845. The raw sequencing data on have been deposited in the bioproject PRJNA823058 (Severe acute respiratory syndrome coronavirus 2 (ID 823058) - BioProject - NCBI (nih.gov)) and under SRA's (https://www.ncbi.nlm.nih.gov/sra) accession codes #SRR19181788-SRR19181796, SRR19181801, SRR191806-SRR19181809, SRR19181812 and SRR19181813.

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

## Acknowledgements

The authors are grateful to the Hemotherapy Service of the Hospital Clementino Fraga Filho (Federal University of Rio de Janeiro, Brazil) for providing buffy coats. We thank Fiocruz and INCa for the use of the cellular and animal biosafety level 3 platforms, respectively. Thanks are also due to the animal biosafety level 3 laboratory at UFMG (Laboratório Institucional de Pesquisa, LIPq); Centro de Laboratórios Multiusuários, CELAM; Laboratório de Biossegurança Nível 3, NB3-ICB. The authors also acknowledge Patricia Machado Rodrigues e Silva Martins and Tatiana Paula Teixeira Ferreira from Fiocruz for the microphotograph analysis. We thank Dr. Ester Sabino and Brazil-UK Center for Arbovirus Discovery Diagnosis Genomics and Epidemiology (CADDE) Genomic Network - Instituto de Medicina Tropical, for donating the P1 VoC. National Institutes of Science and Technology Program (INCT) on Diseases of Neglected Populations (INCT-IDPN, # 465313/2014-0), on Innovation in Medicines and Identification of New Therapeutics Targets (INCT-INOVAMED), on Neuroimmunomodulation (INCT/NIM) and on Dengue and Host-microorganism Interaction (INCT dengue). Universidade Federal de Minas Gerais (UFMG). All authors thank Conselho Nacional de Desenvolvimento Científico e Tecnológico (CNPq, Finance Code 403543/2020-7), Fundação de Amparo à Pesquisa do Estado do Rio de Janeiro (FAPERJ, Finance Code E-26/210.775/2021), Minas Gerais Foundation for Science (FAPEMIG), Coordenação de Aperfeiçoamento de Pessoal de Nível Superior (CAPES) and Empresa Brasileira de Pesquisa e Inovação Industrial (Embrapii; finance code 015/2020). TMLS, JBC, and JR acknowledge the Rede Virus and Dr. Marcelo M. Morales from Ministry of Science, Technology, and Innovation (MCTI).

The work was supported by the Ministry of Science, Technology and Innovation (MCTI) through CNPq (Finance Code 403543/2020-7), FAPERJ (Finance Code E-26/210.775/2021) and Empresa Brasileira de Pesquisa e Inovação Industrial (Embrapii; finance code 015/2020).

## Author contributions

In vitro experiments and analysis: C.Q.S., N.F.R., S.S.G.D., J.R.T.; P.T.B., D.C.B.H., and T.M.L.S.; in vivo experiments and analysis: C.Q.S., A.C.F., M.M., J.V., C.M.Q.-J., L.C.G., I.M.C., P.P.G.G., V.V.C., M.M.T., P.T.B., and T.M.L.S.; enzymatic assays: C.Q.S. and F.P.D.; Molecular Biology experiments: N.F.R., O.A.C., J.R.T., J.G.; coordination of virological studies: T.M.L.S. Synthetic chemistry, design and development of MB-905 and derivatives: V.D.P., P.B.M., A.R.A., and J.A.R. Performed the AMES and micronucleus experiments: C.S.F.; performed the safety pharmacology in vivo and in vitro: S.M.S.-J., S.E.L.T., F.B.P., and A.A.S.; performed pharmacokinetics, toxicokinetic, protein binding and CYP activity: R.M., M.H., and N.F.M.; performed the maximum tolerated dose toxicity study (MTD) and 28-day repeated dose oral toxicity in mice: E.L.A., G.P.F., S.E.L.T., F.B.P., C.F.S., R.M., J.S.-J., and S.M.-J.; coordination, design, implementation and analysis of safe and tolerability studies: JBC; manuscript writing: T.M.L.S., V.D.P., J.B.C., and J.A.R.

## Competing interests

The corresponding authors are among the inventors of the patent requests PCT/BR2021/050136 and PCT/BR2022/050120 on the topic of this study. The remaining authors declare no competing interests.

## Additional information

Article

Thiago Moreno L. Souza [1,2,11] ✉, Vagner D. Pinho [3], Cristina F. Setim[4], Carolina Q. Sacramento[1,2], Rodrigo Marcon[4], Natalia Fintelman-Rodrigues[1,2], Otavio A. Chaves [1,2], Melina Heller[4], Jairo R. Temerozo [1,5,6], André C. Ferreira[1,2,7], Mayara Mattos[1,2], Patrícia B. Momo [3], Suelen S. G. Dias [1], João S. M. Gesto[1,2], Filipe Pereira-Dutra [1], João P. B. Viola[8], Celso Martins Queiroz-Junior[9], Lays Cordeiro Guimarães[10], Ian Meira Chaves[9], Pedro Pires Goulart Guimarães[10], Vivian Vasconcelos Costa[9], Mauro Martins Teixeira[9], Dumith Chequer Bou-Habib [5,6], Patrícia T. Bozza [1], Anderson R. Aguillón[3], Jarbas Siqueira-Junior [4], Sergio Macedo-Junior[4], Edineia L. Andrade[4], Guilherme P. Fadanni [4], Sara E. L. Tolouei[4], Francine B. Potrich[4], Adara A. Santos[4], Naiani F. Marques[4], João B. Calixto [4,11] ✉ & Jaime A. Rabi[3,11] ✉

[1]Laboratório de Imunofarmacologia, Oswaldo Cruz Institute, Fundação Oswaldo Cruz (Fiocruz), Rio de Janeiro, RJ, Brazil. [2]National Institute for Science and Technology on Innovation in Diseases of Neglected Populations (INCT/IDPN), Center for Technological Development in Health (CDTS), Fiocruz, Rio de Janeiro, RJ, Brazil. [3]Microbiológica Química e Farmacêutica, Doutor Nicanor, 238 Inhaúma, Rio de Janeiro, RJ, Brazil. [4]Centro de Inovação e Ensaios Pré-clínicos e National Institute for Science and Technology on Innovation in Medicines and Identification of New Therapeutics Targets (INCT-INOVAMED). Avenida Luiz Boiteux Piazza, 1302 Cachoeira do Bom Jesus, 88056-000 Florianópolis, SC, Brazil. [5]National Institute for Science and Technology on Neuroimmunomodulation (INCT/NIM), Oswaldo Cruz Institute, Fiocruz, Rio de Janeiro, RJ, Brazil. [6]Laboratório de Pesquisa sobre o Timo, Oswaldo Cruz Institute, Fiocruz, Rio de Janeiro, RJ, Brazil. [7]Universidade Iguaçu, Nova Iguaçu, RJ, Brazil. [8]Program of Immunology and Tumor Biology, Brazilian National Cancer Institute (INCA), Rua André Cavalcanti 37, 5th floor, Centro, Rio de Janeiro, Brazil. [9]Centro de Pesquisa e Desenvolvimento de Fármacos, Instituto de Ciências Biológicas, (ICB), Universidade Federal de Minas Gerais (UFMG), Minas Gerais, Brazil. [10]Department of Physiology and Biophysics, Institute of Biological Sciences, Universidade Federal de Minas Gerais, Belo Horizonte, MG, Brazil. [11]These authors contributed equally: Thiago Moreno L. Souza, João B. Calixto, Jaime A. Rabi. ✉e-mail: thiago.moreno@fiocruz.br; joao.calixto@cienp.org.br; jrabi@microbiologica.ind.br

