## [Peer Review File · Nature Communications]

REVIEWER COMMENTS

Reviewer #1 (Remarks to the Author):

Souza and colleagues have presented an in-depth and well designed study assessing the potential of kinetin as an anti-viral, specifically against SARS-CoV-2. It is a thorough evaluation of an antiviral that shows some promise for use. The manuscript would benefit greatly from the addition of subheadings throughout the the results section. This would help the reader better follow along each experiment and the progression of the overall study. I do have a few comments and concerns.

Minor comments:

1. Overall the paper makes some very broad statements I would like to see either referenced or adjusted.

-in the abstract, the line "suggests that mutant viruses might also emerge because of antiviral pressure" is speculative and likely does not belong in the abstract. Antiviral use against SARS-CoV-2 is not yet widespread. As of yet, I have not seen any evidence of this occurring (although molnupiravir sometimes does result in viral recrudescence, the sequence has not changed). This is likely best a discussion point.

-In the abstract, "alleged positive clinical results" should be changed. These results are not alleged, there are several publications showing efficacy of both treatments in animal studies and clinical studies.

-"Continued discovery and development are required as antiviral resistant strains to the first generation of antivirals are likely to emerge" is a very broad statement and not likely applicable to all antiviral compounds.

2. I do not think a YouTube video should be referenced. There must be an official FDA report that can be used instead?

3. The statement "Considering that APRT could convert MB-905 to its riboside 5'-monophosphate and that 5'-nucleotidases (which are common in the liver³⁰, the port of entry of orally available drugs)". I would think the stomach would be the port of entry, I would remove the line "the port of entry from orally available drugs".

4. The modeling of the presence N6-furfuryladenine should be briefly described in the text if possible. There should also be a reference to the "widening of the strand distance" resulting in impaired polymerase activity.

5. Figures 2E and 2F feel like they should be a figure on their own and Figure 2 should contain the data in extended figure 3.

6. MB-905 was found in the plasma and the lungs (Fig 3b and c) please describe the concentrations in the text and relevance to it's antiviral capacity (EC50 etc).

7. There needs to be consistency between the use of the compound name (e.g. MK4482) and the pharmaceutical name (eg Molnupiravir). Please use one name throughout after referencing at the first description.

8. In figure 1 (A) the text reads "or also in the following days" what does this mean. Please define with specific times. Also in fig 1B, please indicate what after 48-72h means, was it different for different compounds? please define.

9. In figure 2, the fonts appear to be different sizes in each panel (particularly C,D and G). Please make consistent.

10. In figure 3, please define metabolite 1-5.

Major concerns:

1. When describing the pathology of the lungs, immunohistochemistry would be very beneficial when comparing the treated vs untreated mice. Also, arrows or some other markings in figure 4 would help better describe the pathology being discussed.

2. In Figure 4 G, is there a way to quantify the pathology (again IHC would help here).

3. Can the authors truly claim anti-inflammatory properties of kinetin in the mouse, or is this a case of less virus = lower immune response in the animals? Although shown in vitro, animals are much more complicated than cell culture and do not translate directly.

4. A twice daily treatment regime should be tried in the mice as well. It would be nice to see the clinical treatment regime tested in the mice. Especially as the half-life of the drug results in compound being below the effective concentration for hours before a new dose is given.

5. Was the viral stock sequenced before use in the mice?

6. The mouse study described in figure does look promising, however why was it only carried out to D7? Would the treated animals have survived if the study had progressed to D21 or even longer? That would be a key finding, especially as the viral titers seem high in the treated animals. Finally, why weren't all the concentrations in study analyzed for viral RNA and infectious titers. It would be nice to see if there is a dose response in the study.

Reviewer #2 (Remarks to the Author):

By testing a library of nitrogenous bases, the study by Souza et al. found that N6-Furfurylaminopurine (kinetin or MB-905), a treatment for familial dysautonomia, inhibits SARS-CoV-2 replication in huh-7 and Calu-3 cells. They also found that MB-905 lowers viral RNA and TNF and IL-6 levels in human donor

monocytes and that the drug synergizes with the HIV integrase inhibitor dolutegravir in Calu-3 cells. Furthermore, they show that MB-905 inhibits viral replication and death in SARS-CoV-2 (P1 strain) infection of the K-18 Ace2-transgenic mouse model. Based on these data and, given that kinetin is less mutagenic than molnupiravir by Ames test, the authors suggest that kinetin could be a new oral treatment for SARS-CoV-2 alone or in combination with dolutegravir. While the biochemical analyses in the study are straight forward, the biological data showing that MB905 inhibits SARS-CoV-2 and its broader effects on inflammation are somewhat preliminary.

Specific Comments:

1. The data in Figure 1A showing SARS-CoV-2 infection of Calu-3 cells and the effect of MB-905 would benefit from being displayed relative to mock for each drug with the cytotoxicity of each compound vs. carrier at each concentration simultaneously shown so that the reader can evaluate the concentration effect of the drug on cytotoxicity.

2. The authors state that 'compounds were added after the viral inoculum was removed and cells washed, they also state, 'or also in the following days'...what does that mean? How many hours after infection? and was it redosed daily? Were all the compounds evaluated simultaneously? Furthermore, in Fig 1B, the authors write that supernatants were harvested after 48-72 hrs. Was it 48 or 72 hrs and were all the compounds evaluated simultaneously or if not, how was the experiment performed?

3. The data in Fig 1C-E show SARS-CoV-2 infection of donor monocytes. While viral load is shown in 1C there is no data of percent infection by the virus of the monocytes or visualization of the infection by for example FACS. This is important to show that infection was robust and inhibited by MB-905 and correlated with the inhibition of viral RNA production. Furthermore, there is no cytotoxicity data shown in the monocytes studies, which is required to make these points. The same controls pertain to interpretation of Figs 1D and 1E. 'Nil' ...appears to be no treatment is that correct? Is DMSO the carrier for each of the compounds? If so, it would be helpful to have it as a control in these measurements and in measurement of cytotoxicity of the monocytes (and Calu-3).

4. The authors state in several places in the manuscript that Calu-3 cells are type II pneumocytes. This is not accurate. They are a transformed cell line derived from a metastatic pleural lesion from a non-small cell adenocarcinoma originating in the lung. While they are a useful human workhorse cell line because they are Ace2+ and can be infected by SARS-CoV-2, and do originate from a transformed pulmonary tumor, they are not a type II human pneumocyte.

5. In Figure 4, the authors show the effect of MB905 +/- dolutegravir on SARS-CoV-2 infection of the Ace2+ transgenic K18 mouse model. Higher survival rates were observed for MB-905 at 140 mg/kg/day, combined or not with DTG, and at 70 mg/kg/day when combined with dolutegravir (Figure 4A). This is

supported by the synergy experiment in Calu-3 cells using synergy -finder. Mitigation of weight loss is also observed after SARS-CoV-2 infection by MB905 140 mg, but best with DTG alone at Day 6. What do the authors make of this? Drug mediated viral inhibition does not appear to be significant on prior days-- the value of the p values should be stated somewhere. I am assuming all of the treatments result in a significant improvement of weight loss--the figure is a bit ambiguous. But in this case DTG alone performs best—how is that explained?

6. BAL viral titers are reduced in K-18 mice infected with SARS-CoV-2 when treated with MB905 +/- DTG. It would be of interest to know if they are lower in lung parenchyma as well.

Pathology is shown (H&E staining of lung parenchyma) and the authors state that “Whereas infected/untreated mice lung histology displays collapsed alveoli septum and intense hemorrhage, treated animals displayed a lung parenchyma closer to mock-infected mice”. While it may be closer from the magnification shown with one example, it seems difficult to come to that conclusion. How many mice were evaluated for each group? It would be helpful to see quantification of these features and also potential inflammatory features in animal lungs using a well-accepted scoring quantification by a pathologist. While TNF, IL-6, KC, were quantified in mouse BAL at one timepoint, the conclusion that MBP-905 has ‘significant anti-inflammatory properties’ is overstated given that lung parenchyma inflammation is not demonstrated. TNF and IL-6 are inducible by SARS-CoV-2 and the lower levels of these cytokines could be secondary to lower viral load and not to MB-905 having a direct and independent anti-inflammatory function.

6. An important finding with respect to MB-905’s potential use as an antiviral for SARS-CoV-2 is the demonstration that although MB-905 leads to error-prone virus replication it was negative for both Ames and micronucleus tests, unlike molnupiravir.

7. While kinetin has been used to treat familial dysautonomia it is important to include an estimate of how many patients with this rare disease have actually been treated and to describe the side effects that have been associated with its use.

REVIEWER COMMENTS

Reviewer #1 (Remarks to the Author):

Souza and colleagues have presented **an in-depth and well designed** study assessing the potential of kinetin as an anti-viral, specifically against SARS-CoV-2. **A thorough evaluation** of an antiviral shows some promise for use. The manuscript would benefit greatly from the addition of subheadings throughout the the results section. This would help the reader better follow along each experiment and the progression of the overall study. I do have a few comments and concerns.

Minor comments:

1. Overall the paper makes some very broad statements I would like to see either referenced or adjusted. -in the abstract, the line "suggests that mutant viruses might also emerge because of antiviral pressure" is speculative and likely does not belong in the abstract. Antiviral use against SARS-CoV-2 is not yet widespread. As of yet, I have not seen any evidence of this occurring (although molnupiravir sometimes does result in viral recrudescence, the sequence has not changed). This is likely best a discussion point.

The Reviewer is correct, and in this revised version, we adjusted some broad statements. With respect to the notion of SARS-CoV-2 antiviral resistance, since it is not a disseminated problem at moment, we removed and adjusted the language in the abstract. Nevertheless, some works have started to report SARS-CoV-2 resistance to remdesivir and nirmatrelvir (<https://www.nature.com/articles/s41467-022-29104-y> and <https://www.biorxiv.org/content/10.1101/2022.06.28.497978v1>); thus, we kept this problem among our arguments in the other sections of the manuscript.

-In the abstract, "alleged positive clinical results" should be changed. These results are not alleged, and there are several publications showing the efficacy of both treatments in animal studies and clinical studies.

The text has been revised accordingly.

-"Continued discovery and development are required as antiviral resistant strains to the first generation of antivirals are likely to emerge" is a very broad statement and unlikely applicable to all antiviral compounds.

This phrase was removed.

2. I do not think a YouTube video should be referenced. There must be an official FDA report that can be used instead?

We added a reference from FDA, but we would like to keep the YouTube link as the primary source of information of the 9 h-meeting that authorized Molnupiravir. As the Journals style allows us to share web

links, we think it is didactical for those interested in understanding the reasons why the FDA committee members decided to rule in or out this drug's authorization.

3. The statement "Considering that APRT could convert MB-905 to its riboside 5'-monophosphate and that 5'-nucleotidases (which are common in the liver³⁰, the port of entry of orally available drugs)". I would think the stomach would be the port of entry, I would remove the line "the port of entry from orally available drugs".

The sentence was removed.

4. The modelling of the presence N6-furfuryladenine should be briefly described in the text if possible. There should also be a reference to the "widening of the strand distance" resulting in impaired polymerase activity.

We have revised the methods section of the in silico analysis to provide more accurate details. To support the notion that N6-furfuryladenine may lead to noncanonical base pairing, in a wobble-like confirmation, we made extended Figure 4 of the revised version more didactic and incorporated the RMSD differences and distances when N6-furfuryladenine was incorporated into at least one of the RNA strands. Our new data show that the furan ring in MB-905 may increase the strand distance by approximately 1.3 Angstroms.

5. Figures 2E and 2F feel like they should be a figure on their own and Figure 2 should contain the data in Extended Figure 3

We followed the Reviewer's suggestion and changed Figure 2 New Figure 2 and incorporated the data from the old extended Figure 3 as panels E and F. The old Figure 2E-G panels became the new Figure 3 in the revised version, specifically describing the displays on the experiments on drug combination. All the numbers of the main and extended figures were adjusted.

6. MB-905 was found in the plasma and the lungs (Fig 3b and c) please describe the concentrations in the text and relevance to it is antiviral capacity (EC₅₀ etc.).

Information on the MB-905 concentration in the plasma and lungs of mice treated with 140 mg/kg and the respective EC₅₀ and EC₉₀ values are presented in the new Figure 5 in the revised version of the manuscript.

7. There needs to be consistency between the use of the compound name (e.g., MK4482) and the pharmaceutical name (e.g., Molnupiravir). Please use one name throughout after referencing at the first description.

We adjusted the text to preferentially use the name monupiravir.

8. In figure 1 (A) the text reads "or also in the following days" what does this mean. Please define with specific times. Additionally, in fig 1B, please indicate what after 48-72 h means, was it different for different compounds? please define.

We made the text clearer by indicating that infected cells were treated just once postinfection or an additional treatment at 24 h post-infection was also used. We have also specified in the Methods section and figure Legends the time that culture supernatant was harvested for titration to be 48 h after infection.

9. In figure 2, the fonts appear to be different sizes in each panel (particularly C,D and G). Please make consistent.

The panels in Figure 2 were revised.

Major concerns:

1. When describing the pathology of the lungs, immunohistochemistry would be very beneficial when comparing the treated vs untreated mice. Additionally, arrows or some other markings in figure 4 would help better describe the pathology being discussed.

In vivo experiments were extensively revised in this newer version of the manuscript. Please see the new section 2.5, which describes that SARS-CoV-2-infected transgenic mice expressing human ACE2 and hamsters were protected by MB-905. Figures 5, 6, 7 and Extended Figure 5 were incorporated in this revision. Representative histological images along with pathological scores and IHC for dsRNA are presented now.

2. In Figure 4 G, is there a way to quantify the pathology (again IHC would help here).

This information was included in the revised manuscript, new Figures 6 and 7.

3. Can the authors truly claim anti-inflammatory properties of kinetin in the mouse, or is this a case of less virus = lower immune response in the animals? Although shown in vitro, animals are much more complicated than cell culture and do not translate directly.

Referee is right. We alleviate the language on an endogenous anti-inflammatory activity for MB-905, and now we correlated it with the drugs' ability to reduce the viral insult.

4. A twice daily treatment regimen should be tried in the mice as well. It would be nice to see the clinical treatment regime tested in the mice. In particular, the half-life of the drug results in the compound being below the effective concentration for hours before a new dose is given.

We performed a twice-daily treatment in hamsters. Our most complete preclinical safety profile is based on toxicological experiments on animals larger than mice. The experimental rats in our study tolerated MB-905 well at 250 mg/kg (Supplementary Figure 3). According to standard conversion calculations between species (<https://www.ncbi.nlm.nih.gov/pmc/articles/PMC4804402/>), 250 mg/kg in rats is equivalent to 296 mg/kg in hamsters. For practical reasons, of drug dilution and combinations, we aimed to use 140 mg/kg twice a day (BID), which results in a daily exposure below the equivalent dose of preclinical support. Both mock- and SARS-CoV-2-infected hamsters tolerated the doses of 140 mg/kg once a day (QD) and BID well (Figure 7A). Similar to infected mice, MB-905 did not affect viral RNA levels (Figure 7B), whereas BID treatment reduced infectious titers (Figure 7C). Consequently, this treatment protected the lungs from infected and treated hamsters (Figure 7D-F).

5. Was the viral stock sequenced before use in the mice?

Yes, they were. The following information was highlighted in the methodology: “The SARS-CoV-2 B.1 lineage (GenBank #MT710714) and gamma variant (also known as P1 lineage; #EPI_ISL_1060902) were isolated on Vero E6 cells from nasopharyngeal swabs of confirmed cases”.

6. The mouse study described in figure does look promising, however why was it only carried out to D7? Would the treated animals have survived if the study had progressed to D21 or even longer? This would be a key finding, especially as the viral titers seem high in the treated animals. Finally, why were not all the concentrations in study analysed for viral RNA and infectious titers. It would be nice to see if there is a dose response in the study.

In Brazil, there was no animal biosafety level 3 (ABSL-3) laboratory before the COVID-19 pandemic. The ABSL-3 multiuser labs implemented during the pandemic had a very tight schedule. For logistical reasons, we had to keep the experiments within 7 days. Nevertheless, to respond to this comment, we arranged a two-week time frame experiment, and the results are presented in Figure 5.

The overwhelmed agenda to use this facility also highlights that the demands for K18-hACE mice were huge. Thus, we had to focus on euthanizing the animals in the most suitable conditions to analyse the viral loads.

Reviewer #2 (Remarks to the Author):

By testing a library of nitrogenous bases, the study by Souza et al. found that N6-furfurylaminopurine (kinetin or MB-905), a treatment for familial dysautonomia, inhibits SARS-CoV-2 replication in huh-7 and Calu-3 cells. They also found that MB-905 lowers viral RNA and TNF and IL-6 levels in human donor monocytes and that the drug synergizes with the HIV integrase inhibitor dolutegravir in Calu-3 cells. Furthermore, they show that MB-905 inhibits viral replication and death in SARS-CoV-2 (P1 strain) infection of the K-18 Ace2-transgenic mouse model. Based on these data and given that kinetin is less mutagenic than molnupiravir according to the Ames test, the authors suggest that kinetin could be a new oral treatment for SARS-CoV-2 alone or in combination with dolutegravir. While the biochemical analyses in the study are straight forwards, the biological data showing that MB905 inhibits SARS-CoV-2 and its broader effects on inflammation are somewhat preliminary.

Specific Comments:

1. The data in Figure 1A show SARS-CoV-2 infection of Calu-3 cells, and the effect of MB-905 would benefit from being displayed relative to mock for each drug with the cytotoxicity of each compound vs. carrier at each concentration simultaneously shown so that the reader can evaluate the concentration effect of the drug on cytotoxicity.

The CC₅₀ values of the compounds are displayed in Extended Table 1. We understood by the Reviewer's comment that it was necessary to display the cytotoxicity curves. Now, in the revised manuscript, they are displayed as the new Extended Figure 3.

2. The authors state that 'compounds were added after the viral inoculum was removed and cells washed, they also state, 'or also in the following days'...what does that mean? How many hours after infection? and was it redosed daily? Were all the compounds evaluated simultaneously? Furthermore, in Fig 1B, the authors write that supernatants were harvested after **48-72** hrs. Was it 48 or 72 hrs and were all the compounds evaluated simultaneously or if not, how was the experiment performed?

We apologize for the imprecise annotation in the submitted version. Now, in the revised manuscript, the text was made clearer by indicating that infected cells were treated just once postinfection or that an additional treatment at 24 h postinfection was also used. We specified in the Methods (Yield reduction assay) and figure Legends the time that culture supernatant was harvested for titration.

3. The data in Fig 1C-E show SARS-CoV-2 infection of donor monocytes. While viral load is shown in 1C there is no data of percent infection by the virus of the monocytes or visualization of the infection by for example FACS. This is important to show that infection was robust and inhibited by MB-905 and correlated with the inhibition of viral RNA production. Furthermore, there are no cytotoxicity data shown in monocyte studies, which is required to make these points. The same controls pertain to interpretation of Figs 1D and 1E. 'Nil' ...appears to be no treatment is that correct? Is DMSO the carrier for each of the compounds? If so, it would be helpful to have it as a control in these measurements and in measurement of cytotoxicity of the monocytes (and Calu-3).

Based on this comment, we understand that the Reviewer requests more information and rationalization in the experiments with monocytes. We believe that the clarifications provided here and the changes throughout the manuscript will alleviate the Referee's concern.

Once infected by SARS-CoV-2, monocytes do not produce infectious virus progeny, but virus can enter the cells and synthesize RNA (PMID: 33277988, 33493287, 34401873, 33880524). This nonpermissive SARS-CoV-2 replication in monocytes is harmful for some patients because viral RNA inside monocytes may trigger proinflammatory processes and disruptive cell death, such as pyroptosis (PMID: 33277988, 33493287, 34401873, 33880524).

Indeed Figure 1C is showing the cell-associated SARS-CoV-2 RNA measured by RT-PCR, which is more sensitive than using antibodies to dsRNA and measuring them by FACS. Monocytes, as well as calu-3 cells and Huh-7 cells used in this study, were infected with defined MOIs. As shown in Figure 1C, the antiviral drug treatments reduced virus replication in a dose-dependent manner.

The drug cytotoxicity for monocytes is included in the new Extended Figure 3.

We used "Nil" throughout the manuscript to indicate that the infected cells did not receive an active molecule. In the original submission, under the Material and Methods Section, Reagents subsection we stated: "For in vitro experiments all small molecule inhibitors were dissolved in 100% dimethylsulfoxide (DMSO) and subsequently diluted at least 10⁴-fold in culture or reaction medium before each assay. The final DMSO concentrations showed no cytotoxicity."

4. The authors state in several places in the manuscript that Calu-3 cells are type II pneumocytes. This is not accurate. They are a transformed cell line derived from a metastatic pleural lesion from a non-small cell adenocarcinoma originating in the lung. While they are a useful human workhorse cell line because they are Ace2+ and can be infected by SARS-CoV-2 and originate from a transformed pulmonary tumor, they are not type II human pneumocytes.

The Reviewer is right, calu-3 cells are not stricto sensu type II pneumocytes, but they have been derived from transformed pulmonary tumors. Nevertheless, Calu-3 cells do recapitulate the type II pneumocytes, cells producing type C surfactant that are destroyed in the course of severe COVID-19 (dois: 10.1093/cid/ciaa410 and [10.1016/S2666-5247\(20\)30004-5](https://doi.org/10.1016/S2666-5247(20)30004-5)).

We rephrased the text for the first mention to calu-3, which now reads “...tumor cell line that recapitulates type II pneumocytes (calu-3)^{31,32}...”. The references mentioned above were included. Throughout the text, we made other changes to avoid the direct correlation between calu-3 and type pneumocytes.

5. In Figure 4, the authors show the effect of MB905 +/- dolutegravir on SARS-CoV-2 infection of the Ace2+ transgenic K18 mouse model. Higher survival rates were observed for MB-905 at 140 mg/kg/day, combined or not with DTG, and at 70 mg/kg/day when combined with dolutegravir (Figure 4A). This is supported by the synergy experiment in Calu-3 cells using a synergy-finder. Mitigation of weight loss was also observed after SARS-CoV-2 infection by MB905 140 mg, but best with DTG alone at Day 6. What do the authors make of this? Drug mediated viral inhibition does not appear to be significant on prior days--the value of the p values should be stated somewhere. I am assuming all of the treatments result in a significant improvement of weight loss--the figure is slightly ambiguous. However, in this case DTG alone performs best—how is that explained?

We thank the reviewer for having detected a mistake in the body weight curve for DTG alone, which was adjusted and consequently the survival curve. In the original version, each value of body weight in the DTG group was divided by the mice’s mass in the previous day – instead of the day zero (before infection). As 25% body weight loss is a criterion for euthanasia, the survival curve was also changed.

6. BAL viral titers are reduced in K-18 mice infected with SARS-CoV-2 when treated with MB905 +/- DTG. It would be of interest to know if they are lower in lung parenchyma as well.

We also corrected this information because titers and viral loads were determined from mouse lung extracts. BAL was used to analyse cytokines and LDH levels in the original version, but now in the revision we also included this information from lung extracts. A proxy of viral load levels in lung parenchyma is now presented in Figure 6I, the IHC for dsRNA.

Pathology is shown (H&E staining of lung parenchyma) and the authors state that “Whereas infected/untreated mice lung histology displays collapsed alveoli septum and intense hemorrhage, treated animals displayed a lung parenchyma closer to mock-infected mice”. While it may be closer from the magnification shown with one example, it seems difficult to come to that conclusion. How many mice were evaluated for each group? It would be helpful to see quantification of these features and potential inflammatory features in animal lungs using a well-accepted scoring quantification by a pathologist. While TNF, IL-6, and KC were quantified in mouse BAL at one timepoint, the conclusion that MBP-905 has ‘significant anti-inflammatory properties’ is overstated given that lung parenchyma inflammation is not demonstrated. TNF and IL-6 are inducible by SARS-CoV-2, and the lower levels of these cytokines could be secondary to lower viral load and not to MB-905 having a direct and independent anti-inflammatory function.

In vivo experiments were extensively revised. Please see the new section 2.5, which describes that SARS-CoV-2-infected transgenic mice expressing human ACE2 and hamsters are protected by MB-905. Figures 5,

6, and 7 and Extended Figure 5 were incorporated in this revision. Representative histological images along with pathological scores and IHC for dsRNA are presented now.

6. An important finding with respect to MB-905's potential use as an antiviral for SARS-CoV-2 is the demonstration that although MB-905 leads to error-prone virus replication, it was negative for both Ames and micronucleus tests, unlike molnupiravir.

Thank you

7. While kinetin has been used to treat familial dysautonomia, it is important to include an estimate of how many patients with this rare disease have actually been treated and to describe the side effects that have been associated with its use.

We have been in discussion with the Brazilian regulatory agency for drugs (ANVISA) to push MB-905 towards clinical development. Because we were trying to go to a phase II clinical trial, they suggested that we should use caution data from patients with familial dysautonomia due to their difficulty swallowing and vomiting episodes. ANVISA indicated that we should perform classical phase I trials of dose escalation and multidose regimens, reaching the dose we discussed in our manuscript. Thus, it could be misleading to include information from syndromic and experimentally treated cohorts in our discussion, as it may be difficult to separate side effects and symptoms of familial dysautonomia.

REVIEWERS' COMMENTS

Reviewer #1 (Remarks to the Author):

Souza et al has responded thoughtfully to all comments and suggestion provided. This study has been greatly strengthened by the additional work and changes.